# Nimbus: Secure and Efficient Two-Party Inference for Transformers

**Zhengyi Li**[1,*], **Kang Yang**[3,†], **Jin Tan**[4], **Wen-jie Lu**[4], **Haoqi Wu**[4], **Xiao Wang**[5],
**Yu Yu**[1,2], **Derun Zhao**[4], **Yancheng Zheng**[4], **Minyi Guo**[1,2], **Jingwen Leng**[1,2,†]

[1]Shanghai Jiao Tong University, [2]Shanghai Qizhi Institute, [3]State Key Laboratory of Cryptology
[4]Ant Group, [5]Northwestern University

## Abstract

Transformer models have gained significant attention due to their power in machine learning tasks. Their extensive deployment has raised concerns about the potential leakage of sensitive information during inference. However, when being applied to Transformers, existing approaches based on secure two-party computation (2PC) bring about efficiency limitations in two folds: (1) resource-intensive matrix multiplications in linear layers, and (2) complex non-linear activation functions like GELU and Softmax. This work presents a new two-party inference framework Nimbus for Transformer models. For the linear layer, we propose a new 2PC paradigm along with an encoding approach to securely compute matrix multiplications based on an outer-product insight, which achieves $2.9\times \sim 12.5\times$ performance improvements compared to the state-of-the-art (SOTA) protocol. For the non-linear layer, through a new observation of utilizing the input distribution, we propose an approach of low-degree polynomial approximation for GELU and Softmax, which improves the performance of the SOTA polynomial approximation by $2.9\times \sim 4.0\times$, where the average accuracy loss of our approach is 0.08% compared to the non-2PC inference without privacy. Compared with the SOTA two-party inference, Nimbus improves the end-to-end performance of $\text{BERT}_{\text{base}}$ inference by $2.7\times \sim 4.7\times$ across different network settings.

## 1 Introduction

Transformer models [36] bring about significant advancements in various machine learning tasks, such as language understanding [19], vision tasks [6], and chatting bot [21]. As Transformer models handle increasingly sensitive data and tasks, privacy becomes a major concern for deployment. For example, one hospital (client) wants to use the model from another organization (server) to enhance its diagnostic capabilities. This raises privacy concerns for both parties: either the hospital has to upload its private data, or the organization needs to send its proprietary model to the hospital.

Recently, several works [14, 26, 16, 29], building upon secure two-party computation (2PC), realize secure two-party inference in a privacy-preserving way. These works proceed by having the client and server jointly execute inference over the "encrypted" input and model, using cryptographic techniques including homomorphic encryption (HE) [7], additive secret sharing, etc. The client learns nothing about the model except for inference results and keeps the server unknown for the client's input.

Privacy protection comes with substantial computation and communication costs due to expensive cryptographic operations. While existing secure two-party inference for convolution neural networks could be completed in a few minutes [8, 23, 18, 2, 34, 32, 17], it is more challenging to make secure

---

[*]Email at hobbit@sjtu.edu.cn

[†]Corresponding authors, email at: yangk@sklc.org and leng-jw@sjtu.edu.cn

38th Conference on Neural Information Processing Systems (NeurIPS 2024).

inference on Transformer models practical, due to heavy matrix multiplications in linear layers and complex non-linear layers. To amortize the overhead of HE in linear layers, many works [17, 14, 26] adopt *window encoding* to simulate the inner product. However, such an encoding approach brings about a *sparse* format of HE ciphertexts, leading to redundant communication and computation. The efficiency bottleneck of non-linear layers is to securely compute GELU and exponential (included in Softmax). Prior works [5, 26] use piecewise polynomials to approximate the two non-linear functions. However, high-degree polynomials and large fixed-point precision are used to maintain the accuracy, which causes large communication costs and rounds.

This work proposes a new secure two-party inference framework Nimbus for Transformer models to address the above efficiency bottlenecks. Specifically, our contributions are summarized as follows:

- We propose a Client-Side Outer Product (COP) protocol to facilitate linear layers. Our COP protocol incorporates two key innovations. First, the static nature of model weights allows the server to send encrypted weights to the client during the setup stage, thereby eliminating input communication during the online stage. Second, removing input communication enables us to design a novel row-wise encoding scheme that achieves homomorphic matrix multiplication via the outer product. Such encoding further enhances the efficiency of homomorphic matrix multiplication and yields compact output ciphertexts for communication.

- For non-linear layers, we present a new observation that their input distribution exhibits regular patterns. Unlike prior approximations that assumed a uniform input distribution, our approach reduces the approximation budget allocated to seldom-occurring input values. This enables us to use lower-degree polynomials and fewer pieces to approximate non-linear functions. Additionally, low-degree polynomials demonstrate lower sensitivity to fixed-point errors, allowing us to adopt a smaller ring for greater efficiency. We also propose a new protocol that enables *free* conversion between the small and large rings. Consequently, our approach achieves improved performance for non-linear layers while incurring only an average accuracy loss of 0.08%.

- We evaluate the performance of Nimbus using the popular Transformer model $\text{BERT}_{\text{base}}$ under both LAN and WAN settings. Compared with the SOTA work BumbleBee [26], we improve the performance of securely computing matrix multiplication (resp., GELU and Softmax) by $2.9\times \sim 12.5\times$ (resp., $2.9\times \sim 4.0\times$). Combining all the optimizations, we improve the end-to-end performance of secure two-party inference by $2.7\times \sim 5.9\times$ and reduce the communication cost by $60\%$. The code is available at: https://github.com/secretflow/spu.

## 2 Background

We present the necessary background, including the threat model, cryptographic building blocks, and secure Transformer inference.

### 2.1 Threat Model

Our protocol works in the two-party setting where the client $C$ holds an input and the server $S$ holds a model. Our protocol is secure in the presence of a semi-honest adversary who could passively corrupt either the client or the server, where the adversary follows the protocol specification but may try to learn more information than allowed. Semi-honest adversary is a common assumption for privacy-preserving machine learning and has been used in most two-party protocols [18, 34, 17, 14]. As in all prior two-party inference protocols, the client is only allowed to learn the model's architecture and inference result while the server gains no information about the client's input.

### 2.2 Notation

We use upper-case bold letters to represent matrices like $\mathbf{W}$ for model weights and $\mathbf{X}$ for activations. For a matrix $\mathbf{W}$, we use $\mathbf{W}_i$ to denote the $i$-th row of $\mathbf{W}$ and $\mathbf{W}_{i,j}$ to denote the entry in the $i$-th row and $j$-th column of $\mathbf{W}$. For an integer $n$, we write $[n] = \{0, 1, \cdots, n-1\}$. For an additive secret sharing $\langle \cdot \rangle$ (defined in Section 2.3), we use $\langle \cdot \rangle_c$ (resp., $\langle \cdot \rangle_s$) to denote a share held by a client (resp., a server). We denote by $[\![\mathbf{M}]\!]$ the homomorphic encryption (HE) ciphertexts on matrix/vector $\mathbf{M}$ where it may consist of multiple ciphertexts. We use $\mathbb{Z}_{2^\ell} = \mathbb{Z} \cap [0, 2^\ell)$ to denote a ring with all

entries modulo $2^\ell$. For a power-of-two integer $N$, we use $\mathbb{A}_{N,2^\ell} = \mathbb{Z}_{2^\ell}[X]/(X^N + 1)$ to denote a set of polynomials over a ring $\mathbb{Z}_{2^\ell}$. Besides, we use lower-case letters with a "hat" symbol such as $\hat{a}$ to represent polynomials of degree $N - 1$ in $\mathbb{A}_{N,2^\ell}$, where $\hat{a}[j]$ denotes the $j$-th coefficient of polynomial $\hat{a}$. Note that a polynomial $\hat{a}$ can encode at most $N$ elements over $\mathbb{Z}_{2^\ell}$.

## 2.3 Building Blocks

Our framework is built upon multi-party computation (MPC) techniques, including additive secret sharings and homomorphic encryption (HE). The building block of oblivious transfer (OT) and the sub-protocols used in non-linear layers can be found in Appendix B.

**Additive Secret Sharings.** In the two-party setting, an additive secret sharing over a ring $\mathbb{Z}_{2^\ell}$ is defined as: for a value $x \in \mathbb{Z}_{2^\ell}$, two random shares $\langle x \rangle_c \in \mathbb{Z}_{2^\ell}$ and $\langle x \rangle_s \in \mathbb{Z}_{2^\ell}$ are sampled uniformly such that $x = \langle x \rangle_c + \langle x \rangle_s \mod 2^\ell$, where $\langle x \rangle_c$ is held by a client and $\langle x \rangle_s$ is held by a server.

**Homomorphic Encryption.** We adopt the lattice-based additive HE scheme [26] (building upon [33]). HE allows one party to perform computations on the encrypted data of the other party without the need for the decryption key. The HE scheme encodes a plaintext vector $\mathbf{m} \in (\mathbb{Z}_{2^\ell})^N$ into a plaintext polynomial $\hat{m} \in \mathbb{A}_{N,2^\ell}$, and then $\hat{m}$ is encrypted to a ciphertext $[\![\mathbf{m}]\!] = (\hat{b}, \hat{a}) \in \mathbb{A}_{N,q}^2$ where $q$ is a ciphertext modulus. Given a ciphertext $[\![\mathbf{m}]\!]$ and a circuit $f$ including only linear operations, one can homomorphically compute another ciphertext $[\![f(\mathbf{m})]\!]$. We refer the reader to Appendix B.1 for details of the HE scheme and its homomorphic operations.

**Conversion between Floating-point Numbers and Ring Elements.** As 2PC and HE usually operate over rings, the floating-point numbers used in Transformers need to be converted into fixed-point numbers in a ring. Given a scale $s \in \mathbb{N}$ (i.e., the length of the fractional part) and a ring $\mathbb{Z}_{2^\ell}$, a floating-point number $x \in \mathbb{R}$ is converted to the approximated fixed-point number by computing $\tilde{x} := \lfloor x \cdot 2^s \rfloor \mod 2^\ell$, and $\tilde{x}$ can be converted back to the approximated $x$ by setting $\tilde{x}/2^s$.

## 2.4 Secure Two-Party Transformer Inference

The details of the Transformer architecture are described in Appendix A. To securely evaluate the model, the input and output of all layers are in the form of additive secret sharing, enabling the arbitrary linkage of different layers despite specific protocols. This work optimizes the protocol of the linear layers, including $\mathsf{Linear_{qkv}}$, $\mathsf{Linear_o}$, $\mathsf{Linear_{h_1}}$, and $\mathsf{Linear_{h_2}}$. We also optimize the protocols for non-linear layers $\mathsf{Softmax}$ and $\mathsf{GELU}$. The activation multiplication in the attention and layer normalization are relatively fast following SOTA studies [28, 26]. We do not give special optimizations and leave them as future works.

## 3 Secure Computation of Linear Layers

We first analyze the efficiency problems of the prior solution in Section 3.1. Then, Section 3.2 presents our client-side outer product (COP) protocol with row-wise encoding. In Section 3.3, we optimize the memory occupation of our COP protocol.

## 3.1 Prior Solution: Server-side Inner Product Protocol

The starting point of this work is the protocol so-called *server-side inner product (SIP)* [17, 14, 26], as shown in Figure 2(a). The inputs of the linear layer are additive secret sharings $\langle \mathbf{X} \rangle_C, \langle \mathbf{X} \rangle_S \in \mathbb{Z}_{2^\ell}^{k \times m}$ held by the client and server. The server also holds the weights $\mathbf{W} \in \mathbb{Z}_{2^\ell}^{m \times n}$.

$$
\begin{aligned}
\hat{x} = \pi_{\mathrm{L}}(\mathbf{X}) &: \hat{x}[imn + (m-1) - j] = \mathbf{X}_{i,j}, \text{ for } i \in [k], j \in [m] \\
\hat{w} = \pi_{\mathrm{R}}(\mathbf{W}) &: \hat{w}[jm + i] = \mathbf{W}_{i,j}, \text{ for } i \in [m], j \in [n]
\end{aligned}
\tag{1}
$$

The values of two activation shares and server weights are encoded into polynomials over $\mathbb{A}_{N,2^\ell}$ using encoding functions $\pi_{\mathrm{L}} : \mathbb{Z}_{2^\ell}^{k \times m} \to \mathbb{A}_{N,2^\ell}$ and $\pi_{\mathrm{R}} : \mathbb{Z}_{2^\ell}^{m \times n} \to \mathbb{A}_{N,2^\ell}$, as shown in Equation (1). The coefficients of the polynomials $\hat{x}$ and $\hat{w}$ that are not defined are set to 0.

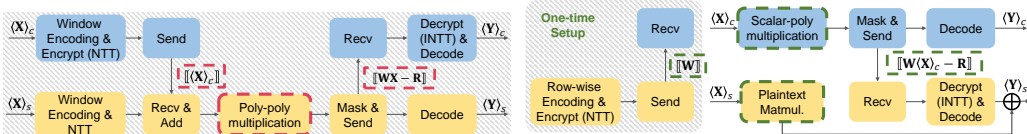

(a) Server-side inner product (SIP) protocol.  (b) Client-side outer product (COP) protocol.

Figure 2: Two rows represent the client and server operations, respectively. The inefficient parts that are accelerated are marked by dashed boundaries. The input communication is shifted as a one-time setup, and the output ciphertexts are compact. The expensive NTT/INTT operations at the online stage are also reduced.

Some of coefficients of polynomial $\hat{z} = \hat{x} \cdot \hat{w} \in \mathbb{A}_{N,2^\ell}$ gives the result of matrix multiplication $\mathbf{Z} = \mathbf{X} \cdot \mathbf{W} \in \mathbb{Z}_{2^\ell}^{k \times n}$, as illustrated in Figure 1. If $kmn > N$, the encoding function would use coefficients with degrees exceeding $N$. The input matrices $\mathbf{X}$ and $\mathbf{W}$ need to be partitioned into smaller windows with respective dimensions $k_w \times m_w$ and $m_w \times n_w$, which results in multiple windows of the output matrix $\mathbf{Z}$ with

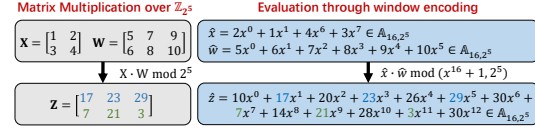

Figure 1: An example of the window encoding of the matrix multiplication using $N = 16$ and $\ell = 5$.

dimension $k_w \times n_w$. Therefore, we refer to this encoding approach as *window encoding*.

Then, the client encrypts her plaintext polynomials as HE ciphertexts $\llbracket \langle \mathbf{X} \rangle_c \rrbracket$, and sends them to the server. After receiving $\llbracket \langle \mathbf{X} \rangle_c \rrbracket$, the server computes the HE ciphertexts $\llbracket \mathbf{X} \rrbracket$ by homomorphically adding $\llbracket \langle \mathbf{X} \rangle_c \rrbracket + \langle \mathbf{X} \rangle_s$. Next, the server homomorphically computes $\mathbf{W} \cdot \llbracket \mathbf{X} \rrbracket - \mathbf{R}$ to obtain HE ciphertexts $\llbracket \mathbf{W}\mathbf{X} - \mathbf{R} \rrbracket$, where $\mathbf{R}$ is randomly generated to mask $\mathbf{Y} = \mathbf{W}\mathbf{X}$ and keeps as the server's output shares $\langle \mathbf{Y} \rangle_s$. Finally, the server sends $\llbracket \mathbf{W}\mathbf{X} - \mathbf{R} \rrbracket$ to the client who decrypts the HE ciphertexts into $\mathbf{W}\mathbf{X} - \mathbf{R}$ that is used as the client's shares $\langle \mathbf{Y} \rangle_c$. Note that number theoretic transform (NTT) [30] is employed to weight plaintext polynomial and activation ciphertext polynomial so that their multiplication complexity is reduced from $O(N^2)$ to $O(N \log N)$.

**Analysis of Communication and Computation Costs.** To simulate the inner product, the window encoding produces a sparse output (e.g. the even-degree terms of $\hat{z}$ in Figure 1). The sparse polynomials are treated as dense after encryption, leading to inefficient communication and computation marked by the dashed boundary in Figure 2(a). First, the computation includes unnecessary zero terms. Second, Iron shows at least $\frac{2\sqrt{kmn}}{\sqrt{N}}$ ciphertexts are transmitted [14]. Then, BumbleBee [26] proposes a packing approach that trades computation for less communication, but the overall latency is similar.

## 3.2 Client-side Outer Product Protocol

To solve the efficiency problems as described above, we propose an alternative *client-side outer product (COP)* protocol. The COP protocol includes two key insights. First, the static nature of model weights allows the server to send encrypted weights to the client at the setup stage, which can eliminate input communication at the online stage. Second, this elimination of input communication enables us to design a new row-wise encoding that realizes homomorphic matrix multiplication through the outer product. Our encoding further results in compact output ciphertext for communication and enhances the efficiency of HE matrix multiplication. The formal protocol is described in Appendix C.1.

**COP Protocol.** In Figure 2(b), we describe the COP protocol for secure matrix multiplication, where the dashed boundary shows the optimizations of computation and communication over prior works. In the setup stage, the server encodes the model weights $\mathbf{W}$ in a row-wise fashion and sends the HE ciphertexts $\llbracket \mathbf{W} \rrbracket$ on these weights to the client. The client stores the HE ciphertexts $\llbracket \mathbf{W} \rrbracket$ in the disk, which enables these ciphertexts to be reused for multiple queries by loading them into memory. In the execution stage, for additive secret sharings $\langle \mathbf{X} \rangle_c, \langle \mathbf{X} \rangle_s$ held by the client and server respectively, the client homomorphically computes $\langle \mathbf{X} \rangle_c \cdot \llbracket \mathbf{W} \rrbracket$ to obtain HE ciphertexts $\llbracket \mathbf{W} \langle \mathbf{X} \rangle_c \rrbracket$, and the server locally computes $\mathbf{W} \cdot \langle \mathbf{X} \rangle_s$ in plaintext. Then, the client samples a random matrix $\mathbf{R}$ (used as its

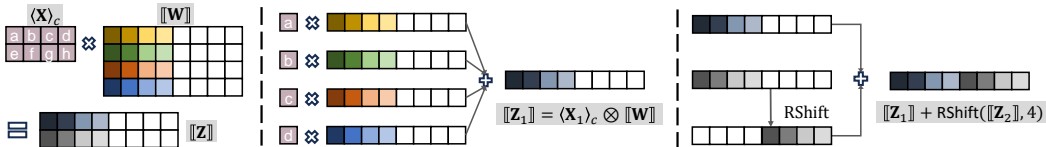

Figure 3: Illustration of our matrix multiplication. **Left:** Functionality of the matrix multiplication using row-wise encoding. **Middle:** Computing the first row of the output through the scalar-poly product. **Right:** Packing two ciphertexts using aright shift for less number of output ciphertext.

output shares $\langle \mathbf{Y} \rangle_c = \mathbf{R}$ where $\mathbf{Y} = \mathbf{WX}$), and homomorphically computes $[\![\mathbf{W}\langle \mathbf{X} \rangle_c]\!] - \mathbf{R}$ to obtain HE ciphertexts $[\![\mathbf{W}\langle \mathbf{X} \rangle_c - \mathbf{R}]\!]$ which is sent to the server. Finally, the server decrypts these ciphertexts to obtain $\mathbf{W}\langle \mathbf{X} \rangle_c - \mathbf{R}$, and then sets its output shares as $\langle \mathbf{Y} \rangle_s = \mathbf{W}\langle \mathbf{X} \rangle_c - \mathbf{R} + \mathbf{W}\langle \mathbf{X} \rangle_s$. As a result, two parties hold additive secret sharings $(\langle \mathbf{Y} \rangle_c, \langle \mathbf{Y} \rangle_s)$. Below, we explain the key process for computing $\langle \mathbf{X} \rangle_c \cdot [\![\mathbf{W}]\!]$ using the row-wise encoding approach.

**Row-wise Encoding.** Assigning the client to perform the plaintext-ciphertext multiplication has a direct benefit in terms of saving the communication of inputs. More importantly, such saving enables us to design a new encoding approach for the weights and activations. First, we use the following function to encode each row of the weight matrix, i.e., $\mathbf{W}_i$ for $i \in [m]$,

$$\hat{w}_i = \pi_R(\mathbf{W}_i) \text{ s.t. } \hat{w}_i[j] = \mathbf{W}_{i,j}, \text{ for } j \in [n]. \tag{2}$$

The output dimensions $n$ of Transformers are generally less than $N$, so the last $N - n$ coefficients are set as zero, indicated by the blank squares in Figure 3. Row-wise encoding corresponds to an extreme case of prior window encoding (i.e., setting $n_w = n$), which may cause a large number of ciphertexts on inputs. This problem is solved by eliminating the input communication in our COP protocol. Second, for the activations, the client no longer encodes his share into polynomials but directly performs multiplication between the plaintext activation shares and ciphertext on weights.

**Efficient Computation and Compact Output Ciphertexts.** Our encoding realizes secure matrix multiplication through outer product, which achieves more efficient homomorphic computation and compact HE ciphertexts on the output. We illustrate the computation of the first row of the output matrix $\mathbf{Z}_1$ in Figure 3. Following the spirit of the outer product, each scalar-polynomial multiplication produces a partial sum of $\mathbf{Z}_1$ and their accumulation produces the final output. The scalar-polynomial multiplication has same complexity as the prior poly-poly multiplication in the NTT space. But we reduce the online NTT/INTT operation, and thus reduce the computation cost. For the output communication, if $n$ is smaller than the polynomial degree $N$, the output ciphertext still leaves blank when communicating. The row-wise encoding makes valid coefficients and zeros separate in the output ciphertext instead of in an interleaved fashion. This enables packing the output ciphertexts through a free operation, right shift. Right shift coefficients for $s$ steps in a ciphertext can be done by multiplying the ciphertext with a plaintext polynomial with only a $s$-order term. The right figure of Figure 3 shows the right shift packing. The second ciphertext is shifted to the right four slots and added with the first ciphertext. Then, all slots of the output are utilized for output communication.

**Complexity Analysis.** Through analysis in Appendix D, Table 1 compares the computation complexity and numbers of communicated ciphertexts. *Communication:* Our COP protocol eliminates the input communication, and output communication is the minimal case of prior $\frac{kn}{k_w n_w}$ when $k_w n_w = N$ since values are densely arranged in the output ciphertext. *Computation:* The SIP protocol requires polynomial-polynomial multiplication. It applies NTT with $O(N \log N)$ complexity to the plaintext polynomial of weights, and ciphertext polynomials of inputs and outputs. Then, the complexity of all poly-poly multiplications in the NTT space is $O(kmn)$. In our protocol, the NTT is only applied when the server decrypts the output ciphertext. The client's scalar-polynomial multiplication saves time spent on the NTT by directly performing the multiplication with a complexity $O(kmN)$, which is comparable to the previous multiplication in the NTT space, especially when $n$ is close to $N$.

### 3.3 Memory Impact of the COP Protocol

In our COP protocol, the client stores the encrypted model weights in the disk so that the ciphertexts can be reused. At the online stage, the encrypted weights are loaded into memory for secure

Table 1: Comparison of the computation and communication for multiplication of two matrices with dimension $k \times m$ and $m \times n$. $k_w, m_w, n_w$ are the window size corresponding to matrix dimensions.

| | SIP | COP |
|---|---|---|
| Communicated Ciphertexts Count | $\frac{km}{k_w m_w} + \frac{kn}{k_w n_w}$ | $k/\lfloor N/n \rfloor$ |
| Server HE Computation Complexity | $O(\frac{mn}{m_w n_w} N \log N + kmn)$ | $O((k/\lfloor N/n \rfloor)N \log N)$ |
| Client HE Computation Complexity | $O\left( \left( \frac{km}{k_w m_w} + \frac{kn}{k_w n_w} \right) N \log N \right)$ | $O(kmN)$ |

matrix multiplication. In this way, the client executes an efficient outer product rather than the original encryption and decryption. The feasibility of such workload reallocation is due to the difference between the "client" in MPC and the traditional client. Due to the symmetric-computation characteristic of MPC as well as the expensive NTT cost brought about by homomorphic encryption and decryption, existing secure-inference frameworks, e.g., [17, 14, 5, 26, 29, 42], require the client to be equipped with similar resources as the server, including a powerful CPU (e.g., 64 vCPUs) and a large memory (e.g., 128 GB) [42, 26, 29]. For clients in MPC, disk usage does not pose a significant issue as storage resources are inexpensive. The CPU usage is also not an issue as the analysis and experiments in Appendix F.2 indicate that the client's computational overhead remains similar as the prior SIP protocol. However, we notice that keeping encrypted weights instead of plaintext weights may introduce additional overhead to memory usage, which we address in the next paragraph.

**Asynchronous Weight Loading.** Our COP protocol allows the weights to be encrypted at the setup stage and stored in the client disk. Different from prior SIP protocol that keeps the model weight shares in memory, loading all encrypted model weights in memory may become a burden since the size of ciphertext is at least four times larger than the secret shares[26]. To reduce the additional usage of memory, we let the client only keep the encrypted weights w.r.t. the current layer in memory (e.g., either 180 MB or 720 MB for Transformer model BERT$_{\text{base}}$). The encrypted weight of the subsequent layer is loaded asynchronously with the communication of output ciphertexts of the linear layer and the secure computation of the following non-linear layer, which involves large-size communication and multiple rounds of interaction. Moreover, the network bandwidth is hundreds of times smaller than the disk bandwidth. For example, as shown in Appendix F.2, loading the encrypted weights of one layer in BERT$_{\text{base}}$ from the disk to memory can be accomplished in tens of milliseconds, while the communication between the client and server requires several seconds. Therefore, the loading time of the encrypted weights can overlap with the communication process.

## 4 Secure Computation of Non-Linear Functions

### 4.1 Prior Solution: Piecewise Polynomial Approximation of Non-Linear Functions

For Transformers, the main efficiency bottleneck in non-linear layers is to securely compute functions exponential and GELU [5, 28, 26]. These works approximate the non-linear functions through piecewise polynomial approximation, which can be securely computed by executing two-party addition, multiplication, and comparison operations. To maintain the accuracy, these works adopt four-piece polynomials with degree 6 for GELU and two-piece Taylor series with Taylor expansion degree 6 for exponential. The approximation of high-degree polynomials inherently imposes a large overhead for securely computing the powers of values. Additionally, such an approximation requires computations to be conducted over a large ring $\mathbb{Z}_{64}$ and with a large scale $s = 18$ [14, 5, 26]. This is brought about by the fact that computing the powers of values with high degrees leads to the accumulation of fixed-point errors and the potential overflow problem.

### 4.2 Simpler Piecewise Polynomial and Smaller Rings by Distribution-aware Approximation

We aim to use simpler piecewise polynomials to fit non-linear functions and reduce the size of rings without sacrificing accuracy. Inspired by the finding that activation distribution exhibits a regular pattern across training and test data [24, 40], our insight for enabling simpler polynomials is to assign the approximation budget according to the input distribution instead of treating all input values with equal importance. Figure 4 illustrates patterns of the input distribution using the BERT$_{\text{base}}$'s

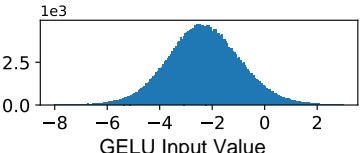 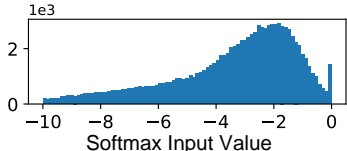

Figure 4: The input distribution of non-linear functions. The y-axis indicates the occurrence counts.

nonlinear functions at the $4_{th}$ encoder. As an example, consider the input distribution of the GELU function. The probability peak centers around $-3$ and values greater than zero occur with less than 10% probability. A wise strategy should leave more budget to the high-probability ranges. Compared with prior research directly minimizing the approximation error of the original function, assuming a uniform input distribution, our strategy is supposed to generate more effective approximations. Additionally, we want to note that the fitted polynomial does not leak the input distribution of the data as the client remains oblivious to the fitted polynomial during secure inference.

**Distribution-aware Splitting and Fitting.** Prior works typically split the input range and fit each interval based on the non-linearity of the curve. We further include the consideration of the input distribution for these two processes. For intervals with low non-linearity or input probability, we split them out and assign constant or linear polynomials to fit. The other intervals with both high non-linearity and input probability are fitted by quadratic or cubic polynomials. When fitting each piece of the non-linear function $f(x)$ by the polynomial $f'(x)$, we minimize the expected loss that integrates the inputs' probability density $p(x)$

$$\min_{f'(x)} \int_l^h p(x) \left[ f(x) - f'(x) \right]^2 dx, \tag{3}$$

where $l$ and $h$ are the lower bound and upper bound. $p(x)$ is the probability density function obtained by summarizing a batch of training data. Unlike prior works using fixed breakpoints $l$ and $h$, we initialize each breakpoint with a starting value and search around it to better fit non-linear functions at different depths. This is because although the activation distributions are broadly similar, they may shift slightly across varying model depths and the breakpoints should be adjusted accordingly. We refer to the Appendix C.2 for the splitting and fitting algorithm. The detailed protocols for securely evaluating nonlinear functions are provided in Appendix C.1. Next, we elaborate on the specific design for fitting exponential and GELU functions.

Exponential. The exponential is used in the Softmax. Given an input vector $\mathbf{x}$, the $i$-th element of Softmax is computed as $\frac{\exp(x_i - \max\{\mathbf{x}\})}{\sum_j \exp(x_j - \max\{\mathbf{x}\})}$. Input values subtracted from maximal values result in maximal zero. The exponential curve exhibits two distinct patterns: a long smooth tail on the left and a sharp increase on the right. Prior works adopt a two-piece approximation by breakpoint -13. Instead, we initial breakpoints around -4 for varying depths. As the right interval spans a smaller range, it adopts a cubic polynomial $P^3(x)$ instead of the Taylor series with expansion degree six [5, 26]. Values less than -4 are less occurred and the curve is smooth, and a linear function is enough to fit.

$$\exp(x) \approx \begin{cases} 0 & x < T_{\exp} \\ P^3(x) & T_{\exp} \le x \le 0 \end{cases} \tag{4}$$

GELU. The GELU curve nearing the zero exhibits pronounced non-linearity. Prior works [5, 26] assign two polynomials for intervals $[-5, -1.97]$ and $[-1.97, 3]$ with degree three and six. We merge these two intervals by one and shrink the range to $[T_1, T_2] = [-2.1, 0.2]$. This is because the values beyond this interval present either less non-linearity or fewer occurrence probabilities, and using constant or linear polynomials is enough. As the middle interval becomes narrow, we find a square polynomial $P^2(x)$ is enough. The specific breakpoints $T_1$ and $T_2$ change for different depths.

$$\text{GELU}(x) \approx \begin{cases} 0 & x \le T_1 \\ P^2(x) & T_1 < x \le T_2 \\ x & x > T_2 \end{cases} \tag{5}$$

Table 2: Accuracy comparison of floating-point (FP) baseline, BumbleBee, Nimbus (without finetuning), and Nimbus[†] (with finetuning).

| Method | CoLA | SST-2 | MRPC | STS-B | QQP | MNLI | QNLI | RTE | Avg. |
|---|---|---|---|---|---|---|---|---|---|
| | Matthews corr. | Acc. | F1 | Pearson | Acc. | Acc. | Acc. | Acc. | |
| FP baseline | 58.63 | 92.88 | 90.12 | 88.24 | 91.22 | 84.74 | 91.28 | 67.87 | 83.12 |
| Bumblebee | 58.40 | 92.88 | 90.12 | 88.28 | 91.21 | 84.74 | 91.39 | 67.87 | 83.11 |
| Nimbus | 58.28 | 92.66 | 89.82 | 87.93 | 90.64 | 84.09 | 90.05 | 66.79 | 82.53 |
| Nimbus[†] | 58.40 | 92.78 | 90.42 | 88.12 | 90.98 | 84.37 | 91.37 | 67.87 | 83.04 |

## 4.3 Free Ring Conversion by Fusion with Truncation

Our low-degree polynomials reduce the errors of operation on fixed-point numbers and potential overflow problems. This enables smaller ring $\mathbb{Z}_{32}$ and precision $s = 12$ for computing Softmax and GELU functions, instead of the original standard ring $\mathbb{Z}_{64}$ and precision of $s = 18$. However, since other operations still require the larger ring to preserve the accuracy, another challenge is to convert secret shares between differently sized rings. The process of downcasting from a larger to a smaller ring can be performed locally, incurring negligible cost [31, 39]. Upcasting from a smaller to a larger ring necessitates addressing the wrap-around of shares, requiring communication among parties. Interestingly, we notice the situations demanding to upcast are always after a truncation operation that inherently computes the wrapping, which can be repurposed for the upcast to avoid additional costs. We propose a novel protocol that fuses the upcast with the truncation. We defer the protocol and the correctness proof to the Appendix E.

## 5 Performance Evaluation

**Experimental Setup.** We follow similar configurations used in prior works [26]. Except optimized non-linear functions using ring $\mathbb{Z}_{2^{32}}$ and precision $s = 12$, other operations follow standard $\mathbb{Z}_{2^{64}}$ and $s = 18$ for the secret sharing. We use $N = 8192$ for the HE encryption. The performances are evaluated on two nodes with 64 vCPUs and 128 GB memory. We use Linux Traffic Control (tc) to simulate LAN and WAN network settings, where the bandwidth and the ping latency are (3Gbps, 1ms) and (400Mbps, 10ms), respectively.

**Baselines.** The baselines include Iron [14] and BumbleBee [26]. Our implementation follows the open-sourced code of BumbleBee on SecretFlow [28]. As Iron is not open-sourced, we implement Iron following their protocol using the SecretFlow library for a fair comparison. For the linear layer, Iron uses window encoding described in Section 3.1, and BumbleBee further compresses the output ciphertext. For non-linear functions, Iron evaluates them via integrating underlying protocols. Later works [5] use piecewise polynomial approximation. BumbleBee further integrates cryptographic optimizations to make a stronger baseline. In the Appendix F.4, we also compare our work with those that use rough approximations to trade off accuracy for efficiency [22, 29].

**Model and Datasets.** Our method is evaluated on widely used Transformer model BERT$_{\text{base}}$ [19] from HuggingFace [38]. When evaluating the performance, we use 128 as a mild average number of the input sequence length. To evaluate the accuracy of our non-linear approximation, we test it on eight datasets from widely used GLUE benchmark [37]. To obtain the input distribution of non-linear functions, we randomly sample sentences from the training dataset until the total token count reaches 512. This number is chosen because further increasing the number of sampled tokens yields no significant changes in the input distribution.

## 5.1 Accuracy Comparison

Table 2 reports the accuracy of floating-point plaintext, BumbleBee, and our approximation across 8 tasks in the GLUE benchmark[37]. The precise approximation of BumbleBee causes small errors due to the truncation error of the fixed-point value computation. Without fine-tuning, Nimbus decreases accuracy in a small range and the average loss is around 0.6%. Such loss can be easily reduced to 0.08% through a lightweight fine-tuning Nimbus[†]. This demonstrates the effectiveness of our

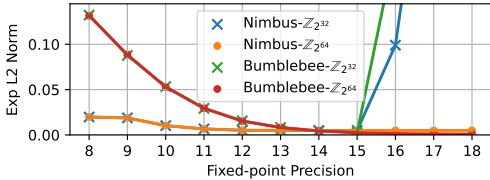 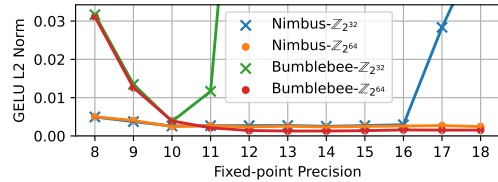

Figure 5: The L2-Norm of output error between oracle non-linear functions and approximations.

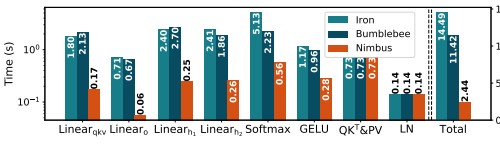

(a) Performance on 3Gbps, 1ms network.

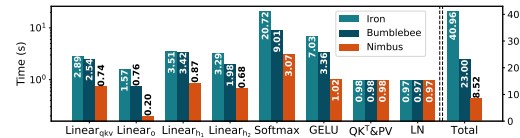

(b) Performance on 400Mbps, 10ms network.

Figure 6: The end-to-end latency of a Transformer block of BERT$_{base}$ and its breakdown.

approximation. We also compare accuracy and efficiency with works that compromise accuracy in Appendix F.4, including MPCFormer [22] and BOLT [29].

Figure 5 presents the output error of exponential and GELU functions to further explain the effectiveness of our approximation. The error is summarized using a batch of test data on a certain layer. On the standard ring $\mathbb{Z}_{2^{64}}$ and precision $s = 18$, our L2-norm errors are around 0.005 and are close to the loss-free approximation of BumbleBee. When reducing to ring $\mathbb{Z}_{2^{32}}$ and $s = 12$, BumbleBee encounters higher errors. This increase is attributed to the more pronounced fixed-point errors that arise when evaluating high-degree polynomials. Moreover, the destructive overflow occurs for precision greater than 10 bits, as the sharp divergence of green and red curves of GELU function. Nimbus has a steady fixed-point error and is not prone to overflow thanks to the low-degree polynomial. This enables moving to the smaller ring with a minor impact on the accuracy.

## 5.2 Efficiency Comparison

As the main body of the Transformer model are identical Transformer blocks, we present the end-to-end latency of one Transformer block under LAN and WAN network settings in Figure 6. Besides the optimized parts of this paper, we unify the unoptimized activation matrix multiplication ($QK^T \& PV$) and LayerNorm (LN) using the BumbleBee's latency. The latencies of Iron are shorter than those reported in their paper due to SecretFlow integrating many SOTA optimizations for the building blocks, such as OT [41] and inverse square root [25]. Combining all our optimizations, the overall runtime is $4.8\times \sim 5.9\times$ faster than Iron and $2.7\times \sim 4.7\times$ faster than BumbleBee. Comprehensive results on varying Transformer size and input sequence length are listed in Appendix F.1. In the following, we provide a detailed analysis of linear and non-linear layers.

For linear layers, our method is efficient in both computation and communication. Therefore, we achieve obvious speedup in both LAN and WAN settings. Compared with stronger BumbleBee, we have $7.2\times \sim 12.5\times$ in the LAN setting and $2.9\times \sim 4.0\times$ in the WAN setting. More speedup for the LAN setting indicates we accelerate the computation more than the communication. For non-linear functions, our method reduces both the communication size and rounds, so that we obtain similar speedup for both the LAN and WAN settings. Compared with stronger baseline BumbleBee, the GELU is $3.4\times$ faster in the LAN setting and $3.3\times$ faster than the WAN setting. The Softmax is $4.0\times$ faster in the LAN setting and $2.9\times$ faster than the WAN setting.

## 5.3 Communication Analysis

Then, we compare the communication cost and the number of rounds of linear layers, Softmax, and GELU in Table 3. The data is summarized using BERT$_{base}$ and sequence length 128. For different types of linear layers, our protocol only requires half the number of communication rounds.

Our total communication size of linear layers is reduced to only 11.51% of that of Iron. Although BumbleBee takes extra "automorphism" operation to compress the output ciphertext, our communication is still only 65% compared with BumbleBee since we also eliminate the communication of the input ciphertext. As for the non-linear layers, compared with stronger baseline BumbleBee, we have fewer rounds and $3\times$ less communication due to simpler piecewise polynomial approximations and the smaller ring size.

Table 3: Communication cost (megabytes) and rounds comparison on one Transformer block.

| Layer | Iron | | BumbleBee | | Nimbus | |
|---|---|---|---|---|---|---|
| | Comm. | Rd | Comm. | Rd | Comm. | Rd |
| $\text{Linear}_{qkv}$ | 74.64 | 2 | 14.47 | 2 | 10.35 | 1 |
| $\text{Linear}_o$ | 40.2 | 2 | 6.71 | 2 | 3.05 | 1 |
| $\text{Linear}_{h_1}$ | 84.46 | 2 | 18.35 | 2 | 15.52 | 1 |
| $\text{Linear}_{h_2}$ | 78.37 | 2 | 15.71 | 2 | 3.05 | 1 |
| Softmax | 689.45 | 110 | 354.26 | 70 | 115.35 | 60 |
| GELU | 283.89 | 65 | 185.13 | 46 | 53.22 | 24 |

# 6 Related Work

**Privacy-preserving Neural Network Inference.** Due to the rapidly growing concerns about data privacy in DNN-based applications, significant efforts have been made to design efficient cryptographic protocols for DNN models [8, 18, 32, 43, 31]. Early works focus on the convolutional neural network (CNN) models. Cryptonets [8] proposed one of the first protocols for 2PC HE-based private neural network inference. Later works [18, 34, 17] are hybrid 2PC neural network inference protocols combining HE for matrix multiplications and multi-party computation for non-linear functions.

**Private Transformers.** Several works have investigated two-party secure inference for the Transformer model. For linear layers, Iron [14] builds upon Cheetah [17] by generalizing the original encoding of matrix-vector multiplication to matrix-matrix multiplication. Both Cheetah and Iron leave blanks in the input and output ciphertexts. BumbleBee [26] utilizes the "automorphism" operation to compress multiple output ciphertexts, which trades computation for communication. A recent work BOLT [29] adopts SIMD encoding to homomorphically evaluate the linear layer, which also trades computation for the compact output ciphertext. All existing works adopt the server-side inner product protocol. In contrast, this work proposes the client-side outer product protocol that eliminates the input ciphertext communication. The proposed protocol also allows a novel encoding approach that facilitates more efficient homomorphic computation and output communication. Other works [1, 5] consider 3PC inference for Transformers, which rely on different settings and cryptographic primitives from this work.

For the non-linear layers, some studies, such as THE-X [3] and MPCFormer [22], evaluate transformer models using cryptographic friendly replacements for non-linear layers, such as using $\text{Softmax} \approx \frac{(\mathbf{x}[i]+c)^2}{\sum_i (\mathbf{x}[i]+c)^2}$ and $\text{GELU}(x) \approx \frac{x^2}{8} + \frac{x}{4} + \frac{1}{2}$. However, such aggressive approximations lead to noticeable accuracy loss, even when employing knowledge distillation to mitigate the decline in accuracy. Other methods, such as look-up tables for faithful approximation [31, 13, 29], are computationally expensive to maintain model accuracy. Later works, including PUMA [5] and BumbleBee [26], utilize piecewise polynomial approximation, which does not result in an accuracy drop but is also relatively costly to compute. In contrast, this work is inspired by insights from the input distribution used in the Transformer model [9, 10, 11, 12, 40, 24]. We propose fitting the non-linear functions according to their input distribution, allowing for lower-degree polynomials and fewer polynomial pieces without sacrificing accuracy.

# 7 Conclusion

We propose a privacy-preserving, accurate, and efficient two-party inference framework Nimbus for Transformers. We present an efficient protocol of secure matrix multiplication using the COP approach, achieving significantly better computation and communication efficiencies. We use a distribution-aware polynomial approximation for non-linear layers, allowing a simpler approximation with less communication and rounds. These optimizations significantly improve the performance, advancing a step towards the practical use of secure Transformer inference.

## Acknowledgement

This work was supported by the National Natural Science Foundation of China grants (62222210, 62102037, 61932019, 92270201, and 62125204). Yu Yu also acknowledges the support from the XPLORER PRIZE. This work was also supported by Ant Group Research Intern Program and we thank all members of the SecretFlow team for their support throughout this project.

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

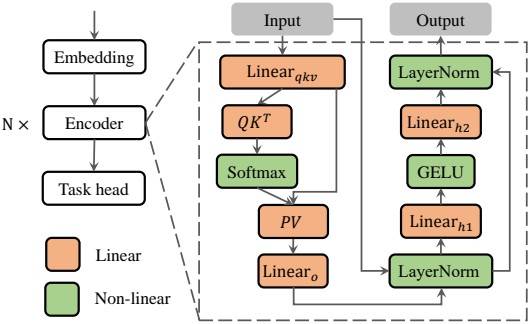

Figure 7: The illustration of the Transformer-based model and the latency breakdown of its private evaluation.

# A  Background of Transformer Models

We focus on Transformer [36] DNN models, such as popular models BERT [19], GPT-2 [4] and LLaMA [35]. These models are stacked with Transformer blocks, each consisting of an attention module and a feed-forward network (FFN) module.

**Attention Module.** The attention module starts with three independent linear layers $\mathsf{Linear}_{qkv}$ that project the input to three activation tensors: $\mathbf{Q}$, $\mathbf{K}$, and $\mathbf{V}$. The multi-head attention mechanism splits them into and computes the self-attention of all heads in parallel through

$$\mathbf{Attention}(Q, K, V) = \mathsf{Softmax}\left(\frac{QK^T}{\sqrt{d_k}}\right)V, \tag{6}$$

where $d_k$ is the hidden dimension of the key activation. The outputs of different heads are concatenated and fed into another linear layer $\mathsf{Linear}_o$, with one residue connection and one normalization layer to generate the final output of the attention module.

**FFN Module.** The FFN module is composed of two linear layers and one activation layer

$$\mathbf{FFN}(\mathbf{X}) = \mathsf{Linear}_{h_2}(\mathsf{GELU}(\mathsf{Linear}_{h_1}(\mathbf{X}))), \tag{7}$$

where GELU [15] is the activation function. Similar to the attention module, its output needs a residue connection and a normalization layer.

**Task Head.** After all the transformer blocks are evaluated, the output is fed into a task-specific head for classification, regression, or token generation.

# B  More Details of Cryptographic Building Blocks

We give a more detailed description of the cryptographic building blocks as a supplement to the paper. We follow those notations used in the paper.

## B.1  Lattice-based Additive Homomorphic Encryption

Homomorphic encryption (HE) enables computation on the encrypted data without knowing the decryption key. This work uses an HE scheme based on Ring Learning-with-Error (RLWE) [27]. The RLWE scheme is defined by a set of public parameters $\{N, q, t\}$, where $N$ is the polynomial degree, $t$ is the modulus of the plaintext, and $q$ is the modulus of the ciphertext.

- **KeyGen.** Generate the RLWE key pair $(sk, pk)$ where the secret key $sk \in \mathbb{A}_{N,q}$ and the public key $\mathsf{pk} \in \mathbb{A}_{N,q}^2$.

- **Encryption.** An RLWE ciphertext is given as a polynomial tuple $(\hat{b}, \hat{a}) \in \mathbb{A}_{N,q}^2$. Given a vector $\mathbf{m}$ which is encoded as $\hat{m} \in \mathbb{A}_{N,t}$, We write $[\![\mathbf{m}]\!] = \mathsf{Enc}(\hat{m})$ to denote the encryption of $\hat{m}$ under a key $pk$.

- **Decryption.** Given an RLWE ciphertext $\llbracket \mathbf{m} \rrbracket = (\hat{b}, \hat{a}) \in \mathbb{A}_{N,q}^2$. We write $\hat{m} = \mathsf{Dec}(\llbracket \mathbf{m} \rrbracket)$ to denote the decryption under a secret key $sk$.

- **Addition ( $\boxplus$ ).** Given two RLWE ciphertexts $\llbracket \mathbf{m}_0 \rrbracket = \left( \hat{b}_0, \hat{a}_0 \right)$ and $\llbracket \mathbf{m}_1 \rrbracket = \left( \hat{b}_1, \hat{a}_1 \right)$ that respectively encrypts $\hat{m}_0, \hat{m}_1 \in \mathbb{A}_{N,t}$ under a same key, the operation $\llbracket \mathbf{m}_0 \rrbracket \boxplus \llbracket \mathbf{m}_1 \rrbracket$ computes the RLWE tuple $\left( \hat{b}_0 + \hat{b}_1, \hat{a}_0 + \hat{a}_1 \right) \in \mathbb{A}_{N,q}^2$ which can be decrypted to $\hat{m}_0 + \hat{m}_1 \mod \mathbb{A}_{N,t}$.

- **Plaintext-ciphertext polynomial Multiplication** ($\boxtimes$)**.** Given an RLWE ciphertext $\llbracket \mathbf{m} \rrbracket = (\hat{b}, \hat{a})$ that encrypts $\hat{m} \in \mathbb{A}_{N,t}$, and a plain polynomial $\hat{c} \in \mathbb{A}_{N,t}$, the operation $\hat{c} \boxtimes \llbracket \mathbf{m} \rrbracket$ computes the tuple $(\hat{b} \cdot \hat{c}, \hat{a} \cdot \hat{c}) \in \mathbb{A}_{N,q}^2$ which can be decrypted to $\hat{c} \cdot \hat{m} \mod \mathbb{A}_{N,t}$.

- **Plaintext-ciphertext scalar-polynomial Multiplication** ($\otimes$)**.** Given an RLWE ciphertext $\llbracket \mathbf{m} \rrbracket = (\hat{b}, \hat{a})$ that encrypts $\hat{m} \in \mathbb{A}_{N,t}$, and a scalar $c$, the operation $c \otimes \llbracket \mathbf{m} \rrbracket$ computes the tuple $(c \cdot \hat{b}, c \cdot \hat{a}) \in \mathbb{A}_{N,q}^2$ which can be decrypted to $c \cdot \hat{m} \mod \mathbb{A}_{N,t}$.

- **Right Shift.** Given an RLWE ciphertext $\llbracket \mathbf{m} \rrbracket = (\hat{b}, \hat{a})$ that encrypts $\hat{m} \in \mathbb{A}_{N,t}$, and a plain polynomial $\hat{c} \in \mathbb{A}_{N,t}$ with a single $s$-order term, the right shift operation $\mathsf{RShift}(\llbracket \mathbf{m} \rrbracket, s)$ computes the tuple $(\hat{b} \cdot \hat{c}, \hat{a} \cdot \hat{c}) \in \mathbb{A}_{N,q}^2$ which can be decrypted to negacyclicly right shift $\hat{m}$ for $s$ terms. This can be implemented through a simple rearrange of the coefficients of the ciphertext, which is a free operation.

## B.2 Oblivious Transfer

OT lets a sender input two messages $m_0, m_1$ and a receiver input a bit $b$, and then the receiver obtains the message $m_b$. For security, the sender is unknown for $b$ and the receiver does not learn $m_{1-b}$. We adopt OT to construct secure two-party computation (2PC) protocols of some non-linear operations such as comparison. We instantiate OT with the communication-efficient Ferret protocol [41].

## B.3 Sub-Protocols for Non-linear Layers

Our protocol for non-linear layers calls the following functionalities in a black-box way to compute element-wise multiplication, comparison, Boolean to Arithmetic share (B2A), and wrap. These functionalities can be securely realized using the known protocols. We use $\langle \cdot \rangle^\ell$ to denote arithmetic additive sharings over a ring $\mathbb{Z}_{2^\ell}$ and $\langle \cdot \rangle^B$ to denote Boolean additive sharings over a binary field $\mathbb{F}_2$. Secret sharing without superscript indicates using the default ring size (e.g., $\mathbb{Z}_{2^{64}}$ for the linear layers and $\mathbb{Z}_{2^{32}}$ for the exponetial and GELU functions).

| Functionalities | Protocols |
|---|---|
| $\langle \mathbf{X} \cdot \mathbf{Y} \rangle = \mathcal{F}_{\mathrm{mul}}(\langle \mathbf{X} \rangle, \langle \mathbf{Y} \rangle)$ | Element-wise multiplication [26] |
| $\langle \mathbf{1}\{x < y\} \rangle^B = \mathcal{F}_{\mathrm{less}}(\langle x \rangle, \langle y \rangle)$ | Less-then [28] |
| $\langle x \rangle^\ell = \mathcal{F}_{\mathrm{B2A}}^\ell(\langle x \rangle^B)$ | Boolean to Arithmetic share [28] |
| $\langle \mathbf{1}\{\langle x \rangle^\ell + \langle y \rangle^\ell \geq 2^\ell\} \rangle^B = \mathcal{F}_{\mathrm{wrap}}(\langle x \rangle^\ell, \langle y \rangle^\ell)$ | Wrap the ring $\mathbb{Z}_{2^\ell}$ [28] |

# C Detailed Protocols and Algorithm

## C.1 Protocols

We present our detailed protocol of matrix multiplication of Section 3 in Algorithm 1. The detailed protocols of securely evaluating GELU and exponential in Section 4 are presented in Algorithm 2 and Algorithm 3.

**Security Proof of the Matrix Multiplication Protocol** The proposed client-side outer product protocol directly builds upon secure building blocks. It guarantees the same security as the traditional server-side inner product protocol. In the presence of a semi-honest adversary, we provide a brief proof idea below. The notations follow those in Algorithm 1.

Specifically, the model weights are encrypted by the server and then sent to the client, where the ciphertexts are denoted by $[\![\mathbf{W}]\!]$. According to the security property of the HE scheme, the ciphertexts reveal no information about these model weights. For secure matrix multiplication, the input matrix $\mathbf{X}$ has been shared as $(\langle \mathbf{X} \rangle_c, \langle \mathbf{X} \rangle_s)$ using additive secret sharing. The client samples a matrix of random shares $\mathbf{R}$, and then homomorphically computes a ciphertext $[\![\langle \mathbf{X} \rangle_c * \mathbf{W} - \mathbf{R}]\!] = \langle \mathbf{X} \rangle_c * [\![\mathbf{W}]\!] - \mathbf{R}$. Due to the circuit-privacy property of the HE scheme, in the client view, the ciphertext $[\![\langle \mathbf{X} \rangle_c * \mathbf{W} - \mathbf{R}]\!]$ does not reveal information on $\mathbf{W}$. This ciphertext is sent to the server, who decrypts it to $\langle \mathbf{X} \rangle_c * \mathbf{W} - \mathbf{R}$. Because of the random mask $\mathbf{R}$, the server also learns nothing about the client share $\langle \mathbf{X} \rangle_c$. In the proof of security, the simulator can simulate the HE ciphertexts using "dummy" ciphertexts on zero, and the adversary's view between the real-world execution and ideal-world execution is proven to be computationally indistinguishable by reducing it to the circuit-privacy security of the HE scheme.

**Security Proof of Non-linear Function Protocols** This work does not modify the protocol for evaluating piecewise polynomials but improves the generation method for these polynomials, ensuring that security remains consistent with previous work. We highlight the training data information that are utilized for fitting non-linear functions is not leaked. This is because the secure evaluation of the piecewise polynomial prevents the client from learning the coefficients and comparison thresholds, as indicated in Algorithm 2 and Algorithm 3. Except for the usage of $\mathbf{b}_0$, which requires the primitive $\mathcal{F}_{\text{mul}}$, all other coefficients are used through addition, which can be performed locally on the server side. For the comparison threshold, the server can subtract the threshold from its share and compare the resulting shares with zero. As a result, the client learns nothing about the piecewise polynomial. A possible improvement on the efficiency is to make $\mathbf{b}_0$ public, thereby saving one round of communication. The only leakage of $\mathbf{b}_0$ does not necessarily lead to leakage of the meaningful information and can provide approximately an 8% speedup.

---

**Algorithm 1** Secure Matrix Multiplication Protocol of Nimbus

**Parties:** $C$ is the client. $S$ is the server owning the model.

**Input:** The client holds activation share $\langle \mathbf{X} \rangle_c \in \mathbb{Z}_{2^\ell}^{k \times m}$. The server holds activation share $\langle \mathbf{X} \rangle_s \in \mathbb{Z}_{2^\ell}^{k \times m}$, $\mathbf{W} \in \mathbb{Z}_{2^\ell}^{m \times n}$, and secret key $sk$.

**Output:** Sharing $\langle \mathbf{Y} \rangle_c \in \mathbb{Z}_{2^\ell}^{k \times n}$ and $\langle \mathbf{Y} \rangle_s \in \mathbb{Z}_{2^\ell}^{k \times n}$ such that $\mathbf{Y} = WX \bmod 2^\ell$.

{Setup phase}

1: Server $S$ partitions the matrix $\mathbf{W}$ into rows $\mathbf{W}_\beta \in \mathbb{Z}_\ell^{1 \times n}$. Then $S$ encodes each row as a polynomial $\hat{w}_\beta = \pi_w(\mathbf{W}_\beta)$ for $\beta \in [m]$. After that $S$ sends $[\![\mathbf{W}_\beta]\!] = \mathsf{Enc}(\hat{w}_\beta)$ for $\beta \in [m]$ to the client $C$.

{Execution phase}

2: The client computes the scalar-polynomial multiplication to obtain a vector of output ciphertexts $\mathbf{c} = [[\![\mathbf{c}_0]\!], [\![\mathbf{c}_1]\!] \cdots [\![\mathbf{c}_{k-1}]\!]]$, where

$$\mathbf{c}[\alpha] = \boxplus_{\beta \in [m]} (x_{\alpha, \beta} \otimes [\![\mathbf{W}_\beta]\!]).$$

for $\alpha \in [k]$. The $\mathbf{c}[\alpha]$ denotes the $\alpha$-th element of the vector $\mathbf{c}$.

3: To compress the the $k$ ciphertexts vector of $\mathbf{c}$ into $k/\lfloor N/n \rfloor$ ciphertexts, The client applies right shift on ciphertexts of $\mathbf{c}$. For example

$$\tilde{\mathbf{c}}[\theta] = \mathrm{RShift}(\mathbf{c}[\theta \cdot \lfloor N/n \rfloor], 0) + \mathrm{RShift}(\mathbf{c}[\theta \cdot \lfloor N/n \rfloor + 1], k) + \cdots$$
$$+ \mathrm{RShift}(\mathbf{c}[\theta \cdot \lfloor N/n \rfloor + \lfloor N/n \rfloor - 1], k * (\lfloor N/n \rfloor - 1))$$

for $\theta \in [k/\lfloor N/n \rfloor]$. Pad with zeros if $k$ cannot be exactly divided by $\lfloor N/n \rfloor$.

4: The client $C$ generates a random polynomial vector $\mathbf{r} = [\hat{r}_0, \hat{r}_0, \cdots \hat{r}_{k/\lfloor N/n \rfloor - 1}]$ to mask the ciphertext. The client sends $\tilde{\mathbf{c}}[\theta] \boxminus \mathbf{r}[\theta]$ to the server for all $\theta$, which are then decrypted by server to obtain $\mathbf{W}\langle \mathbf{X} \rangle_c - \mathbf{R}$. The client keeps $\mathbf{r}$, which is $\mathbf{R} \in \mathbb{Z}_{2^\ell}^{k \times n}$.

5: The server locally computes $\mathbf{W}\langle \mathbf{X} \rangle_s$ and outputs $\langle \mathbf{Y} \rangle_s = \mathbf{W}\langle \mathbf{X} \rangle_s + \mathbf{W}\langle \mathbf{X} \rangle_c - \mathbf{R}$. The client outputs $\langle \mathbf{Y} \rangle_c = \mathbf{R}$.

---

---

**Algorithm 2** Secure GELU Protocol of Nimbus

**Parties:** $C$ is the client. $S$ is the server owning the model. The polynomial $P^2(x)$ with coefficients $\{b_0, b_1, b_2\}$ from Equation 5.

**Input:** The client holds activation share $\langle \mathbf{X} \rangle_c \in \mathbb{Z}_{2^\ell}^{k \times m}$ and the server holds activation share $\langle \mathbf{X} \rangle_s \in \mathbb{Z}_{2^\ell}^{k \times m}$.

**Output:** Sharing $\langle \mathbf{Y} \rangle_c \in \mathbb{Z}_{2^\ell}^{k \times m}$ and $\langle \mathbf{Y} \rangle_s \in \mathbb{Z}_{2^\ell}^{k \times m}$ such that $\mathbf{Y} = GELU(\mathbf{X})$.

1: Two parties locally compute $\langle \mathbf{A}_1 \rangle = \mathcal{F}_{\text{mul}}(\langle \mathbf{b}_0 \rangle, \langle \mathbf{X} \rangle) + b_1$. Then two parties jointly compute $\langle \mathbf{A}_2 \rangle = \mathcal{F}_{\text{mul}}(\langle \mathbf{A}_1 \rangle, \langle \mathbf{X} \rangle) + b_2$. The truncations are implicitly called.

2: Jointly compute the comparisons for interval selection

$$
\begin{aligned}
\langle \mathbf{b}_0 \rangle^B = \mathcal{F}_{less}(\langle \mathbf{X} \rangle, T_1) \quad &\triangleright \mathbf{b}_0 = \mathbf{1}\{\mathbf{X} < T_1\} \\
\langle \mathbf{b}_1 \rangle^B = \mathcal{F}_{less}(T_2, \langle \mathbf{X} \rangle) \quad &\triangleright \mathbf{b}_1 = \mathbf{1}\{T_2 < \mathbf{X}\}
\end{aligned}
$$

$\mathbf{1}\{P\}$ is 1 when the condition $P$ is true and 0 otherwise. Two parties locally set $\langle \mathbf{z}_0 \rangle^B = \langle \mathbf{b}_0 \rangle^B$, $\langle \mathbf{z}_1 \rangle^B = \langle \mathbf{b}_0 \rangle^B$ xor $\langle \mathbf{b}_1 \rangle^B$ xor $l$, $\langle \mathbf{z}_2 \rangle^B = \langle \mathbf{b}_2 \rangle^B$, where $l$ is the party index. In this way, two parties have $\mathbf{z}_0 = \mathbf{1}\{\mathbf{X} < T_1\}$, $\mathbf{z}_1 = \mathbf{1}\{T_1 \leq \mathbf{X} < T_2\}$, and $\mathbf{z}_2 = \mathbf{1}\{T_2 \leq \mathbf{X}\}$.

3: Jointly compute the multiplexing $\langle \mathbf{Y} \rangle = \langle \mathbf{z}_0 \rangle^B \cdot 0 + \langle \mathbf{z}_1 \rangle^B \cdot \langle A_2 \rangle + \langle \mathbf{z}_2 \rangle^B \cdot \langle \mathbf{X} \rangle$ as the output share of each party.

---

**Algorithm 3** Secure Exponential Protocol of Nimbus

**Parties:** $C$ is the client. $S$ is the server owning the model. The polynomial $P^3(x)$ with coefficients $\{b_0, b_1, b_2, b_3\}$ from Equation 4.

**Input:** The client holds activation share $\langle \mathbf{X} \rangle_c \in \mathbb{Z}_{2^\ell}^{k \times m}$ and the server holds activation share $\langle \mathbf{X} \rangle_s \in \mathbb{Z}_{2^\ell}^{k \times m}$.

**Output:** Sharing $\langle \mathbf{Y} \rangle_c \in \mathbb{Z}_{2^\ell}^{k \times m}$ and $\langle \mathbf{Y} \rangle_s \in \mathbb{Z}_{2^\ell}^{k \times m}$ such that $\mathbf{Y} = exp(\mathbf{X})$.

1: Two parties locally compute $\langle \mathbf{A}_1 \rangle = \mathcal{F}_{\text{mul}}(\langle \mathbf{b}_0 \rangle, \langle \mathbf{X} \rangle) + b_1$. Then two parties jointly compute $\langle \mathbf{A}_2 \rangle = \mathcal{F}_{\text{mul}}(\langle \mathbf{A}_1 \rangle, \langle \mathbf{X} \rangle) + b_2$ and $\langle \mathbf{A}_3 \rangle = \mathcal{F}_{\text{mul}}(\langle \mathbf{A}_2 \rangle, \langle \mathbf{X} \rangle) + b_3$. The truncations are implicitly called.

2: Jointly compute the comparisons for interval selection

$$
\langle \mathbf{z}_0 \rangle^B = \mathcal{F}_{less}(\langle \mathbf{X} \rangle, T_{exp}) \quad \triangleright \mathbf{z}_0 = \mathbf{1}\{\mathbf{X} < T_{exp}\} .
$$

$\mathbf{1}\{P\}$ is 1 when the condition $P$ is true and 0 otherwise.

3: Jointly compute the multiplexing $\langle \mathbf{Y} \rangle = (1 - \langle z_0 \rangle^B) \cdot 0 + \langle z_0 \rangle^B \cdot \langle \mathbf{A}_3 \rangle$ as the output share of each party.

---

### C.2 Fitting Algorithm for Non-linear Approximation.

In this section, we present the algorithm used to search the interval breakpoint of the piecewise polynomial. We use the exponential with only one breakpoint as an example to explain. A similar algorithm can be easily generated to the GELU with two breakpoints.

The first step generates the breakpoint candidate set $S$ given the initial breakpoint $T$. One can choose the search range and step according to the needs (Line 1). Then, for each breakpoint candidate, the input range is separated into two intervals (Lines 3-4). We fit both intervals using Equation 3. The required input distribution $p(x)$ can be drawn from a batch of data from the training dataset. The corresponding loss is accumulated for all intervals (Lines 5-9). Then, we update the optimal piecewise approximation (lines 10-13). Finally, the optimal approximation $f'(x)$ is returned.

## D  Complexity Analysis of Linear-Layer Protocol of Nimbus

This section analyzes the computation and communication complexities listed in Table 1. We first analyze the number of HE ciphertexts to be communicated. The SIP protocol requires $\frac{km}{k_w m_w} + \frac{kn}{k_w n_w}$ for the communication of input and output, as we have explained in Section 3.1. Our COP protocol removes the overhead of sending the input $\frac{km}{k_w m_w}$. The scalar-polynomial product produces $k$ output ciphertext, which we pack as $k / \lfloor N/n \rfloor$.

**Algorithm 4** Searching piecewise polynomial approximation of the activation function

---

**Input:** Activation function $f(x)$, initial value of the interval breakpoint $T$, input distribution $p(x)$, and function template of $f'(x)$.
**Output:** The approximated function $f'(x)$;
  1: Generate breakpoint candidates set $S$ around $T$.
  2: Set $best\_loss \leftarrow \infty$ and $f'(x) \leftarrow None$.
  3: **for** $s \in S$ **do**
  4:     Partition the input range into two intervals using $s$.
  5:     $L_{total} = 0$.
  6:     **for** each interval $i$ **do**
  7:         Fit a polynomial of given degree using Equation 3 and obtain the corresponding loss $L_i$.
  8:         Compute total loss $L_{total} += L_i$.
  9:     **end for**
10:     **if** $L_{total} < best\_loss$ **then**
11:         $best\_loss \leftarrow L_{total}$
12:         $f'(x) \leftarrow$ current approximation
13:     **end if**
14: **end for**
**Output:** $f'(x)$

---

**Algorithm 5** Secure Fused Truncation and Upcast.

---

**Input:** Client $C$ and server $S$ hold input $\langle x \rangle^{\ell}$.
**Output:** Client $C$ and server $S$ hold output $\langle y \rangle^{\ell'}$ that $y = x/2^s$.
  1: $S\&C$ invoke $\mathcal{F}_{\text{Wrap}}\left(\langle x \rangle_S^{\ell}, \langle x \rangle_C^{\ell}\right)$ and learn $\langle w \rangle^B$.
  2: $S\&C$ invoke $\mathcal{F}_{\text{B2A}}^{\ell'-\ell+s}\left(\langle w \rangle^B\right)$ and learn $\langle w \rangle^{\ell'-\ell+s}$.
  3: For $b \in \{S, C\}$, $P_b$ outputs $\langle y \rangle_b^{\ell'} = (\langle x \rangle_b^{\ell} >> s) - 2^{\ell-s} * \langle w \rangle_b^{\ell'-\ell+s} \bmod 2^{\ell'}$.

---

Then, we explain the computation complexity. The server in the SIP protocol needs to apply NTT to weight $O(\frac{mn}{m_w n_w} N \log N)$ and the dyadic product $\frac{kmn}{k_w m_w n_w} * N = kmn$. In our scheme, the server only decrypts the $k/\lfloor N/n \rfloor$ output ciphertexts with complexity $O\left((k/\lfloor N/n \rfloor)N \log N\right)$. The client of the SIP protocol needs to perform NTT when encrypting the activation and INTT when decrypting the output, which requires $O\left((\frac{km}{k_w m_w} + \frac{kn}{k_w n_w})N log N\right)$ complexity. In our protocol, the client can directly multiply her activation share with the ciphertext on model weights. Our method has $O(N)$ complexity for each plaintext-ciphertext scalar-polynomial multiplication and $km$ times product with total complexity $O(kmN)$.

## E   Correctness of Truncation-upcast Fusion

The protocol that computes truncation and upcast is in Algorithm 5. We show the correctness of it through the following derivation. Let $\langle x \rangle_i^{\ell}$ ($i \in 0, 1$) denote the secret share held by the client and server on the ring $\mathbb{Z}_{2^{\ell}}$. The second line is drawn from the truncating secret shares on the ring $2^{\ell}$. $w$ is a boolean value indicates the wrap of $\langle x \rangle_i^{\ell}$ over ring size $2^{\ell}$ and $w'$ is the carry bits of the lower $s$ bits. The carry bit $w'$ is either zero or one and can be safely ignored in the inference while the $w \cdot 2^{\ell-s}$ is a significant error that needs to be carefully handled. The third line holds since $\sum_{i=0}^{1} \langle x \rangle_i^{\ell}/2^s - w \cdot 2^{\ell-s} + \hat{w}$ falls within the larger ring $2^{\ell'}$. The fourth line is the modulo expansion of the wrap $w$ on a ring with $k$ bits, where $v$ is a boolean value that indicates the wrap of the $\langle w \rangle^k$. Through a proper choice of $k \geq \ell' - \ell + s$ to promote the boolean share $\langle w \rangle^B$, its wrap can be eliminate by modulo $2^{\ell'}$. The final line indicates the overhead of truncation and upcast is the same as truncation alone, which only requires computing the $\langle w \rangle^k = \langle w \rangle^{\ell'-\ell+s}$.

$$\hat{x}^{\ell'} = x^{\ell}/2^s$$

$$= \sum_{i=0}^{1} \langle x \rangle_i^{\ell}/2^s - w \cdot 2^{\ell-s} + \hat{w}$$

$$= \left( \sum_{i=0}^{1} \langle x \rangle_i^{\ell}/2^s - w \cdot 2^{\ell-s} + \hat{w} \right) \bmod 2^{\ell'} \tag{8}$$

$$= \sum_{i=0}^{1} \langle x \rangle_i^{\ell}/2^s \bmod 2^{\ell'} - \left( \sum_{i=0}^{1} \langle w \rangle_i^{k} - v^{B} \cdot 2^{k} \right) \cdot 2^{l-s} \bmod 2^{\ell'} + \hat{w} \bmod 2^{\ell'}$$

$$\overset{k \geq \ell'-\ell+s}{=} \sum_{i=0}^{1} \langle x \rangle_i^{\ell}/2^s \bmod 2^{\ell'} - \left( \sum_{i=0}^{1} \langle w \rangle_i^{k} \right) \cdot 2^{l-s} \bmod 2^{\ell'} + \hat{w} \bmod 2^{\ell'}$$

# F    More Experiments

## F.1    Comprehensive Performance Comparison

We present a comprehensive end-to-end latency comparison of a transformer block in Figure 8, Figure 9 and Figure 10. We present three model sizes: 768, 1024, and 2048. The input sequence lengths include 1, 32, and 128. Two network conditions are considered: 3000Gbps, 1ms (LAN) and 400Mbps, 10ms (WAN). The sequence lengths of 32 and 128 correspond to the classification Transformer model or the prefill phase of the generative Transformer model. The sequence length 1 can be viewed as the performance during the generation phase of the generative Transformer model. Overall, for the stronger baseline BumbleBee, our method outperforms by a magnitude of $1.9\times$ to $7.6\times$ on sequence lengths 32 and 128. The speedup is minor on sequence 1 with $1.2\times$ to $2.1\times$. For the linear layers, the speedup ranges from $2.47\times$ to $12.09\times$, and the non-linear layers range from $2\times$ to $3.9\times$ In the following, we provide a detailed analysis of the speedup in linear and non-linear layers under varying conditions.

**Linear Layers.** Our method is efficient in both computation and communication. Therefore, we achieve apparent speedup in both LAN and WAN settings. Our method obtains more speedup for large input sequences and hidden dimensions. This is because the computation time of NTT is more dominant for large-size matrix multiplication and input sequences. When the communication speedup is similar, our method has more speedup when the computation time speedup is more significant. Our method has more speedup for the LAN than WAN, where latency is mainly composed by the computation. As our method computes much faster, our latency benefits more from the improved network condition.

**Non-Linear Layers.** Our method is $4\times$ to $10\times$ faster than Iron and $2.0\times$ to $3.9\times$ faster than the stronger baseline BumbleBee. Our speedup on LAN and WAN are similar as our method improves the communication size and the communication rounds. The speedup is also similar for varying sequence lengths and hidden sizes. This is because our optimization lies in a lower degree of approximated polynomials and is not correlated with the input size.

## F.2    Detailed Client Burden Analysis

This section provides evidence to support Section 3.3, including the client computation time and the asynchronous weight loading.

**Client Computation Time Comparison.** The client's computation cost for our COP protocol is similar to the previous SIP protocol. This is based on a counter-intuitive workload distribution of the HE multiplication. Due to the introduction of the NTT, the server in SIP protocol only computes the dyadic product with $O(N)$ complexity, and the more expensive NTT with complexity $O(N \log N)$ is performed by the client when encrypting and decrypting, as listed in Table 1. Our COP protocol makes the client perform a more efficient scalar-vector product with complexity $O(kmN)$. Since

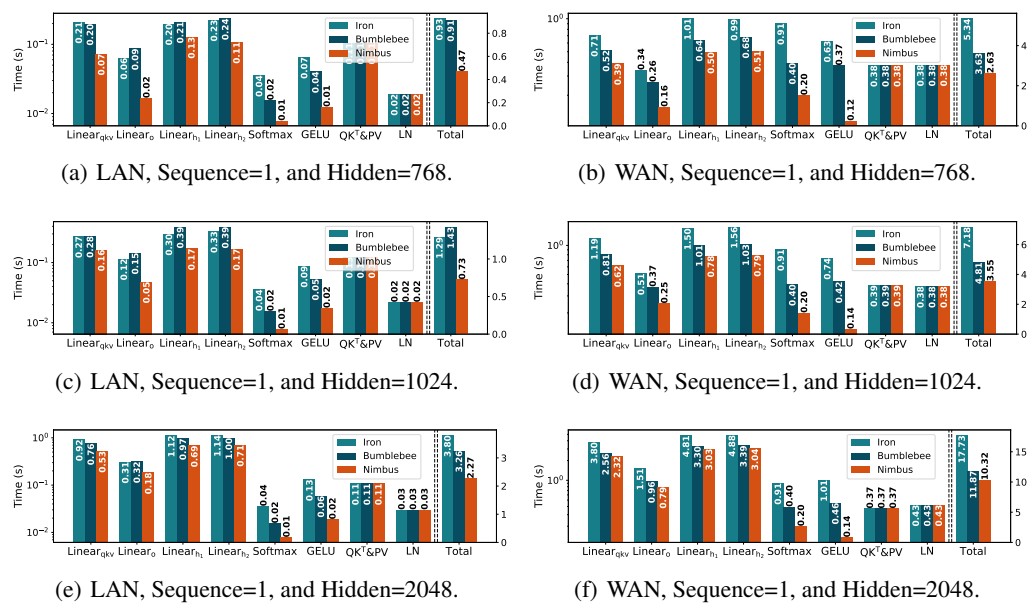

(a) LAN, Sequence=1, and Hidden=768.

(b) WAN, Sequence=1, and Hidden=768.

(c) LAN, Sequence=1, and Hidden=1024.

(d) WAN, Sequence=1, and Hidden=1024.

(e) LAN, Sequence=1, and Hidden=2048.

(f) WAN, Sequence=1, and Hidden=2048.

Figure 8: Under sequence length 1, the end-to-end speedup and breakdown of varying hidden dimensions and network conditions.

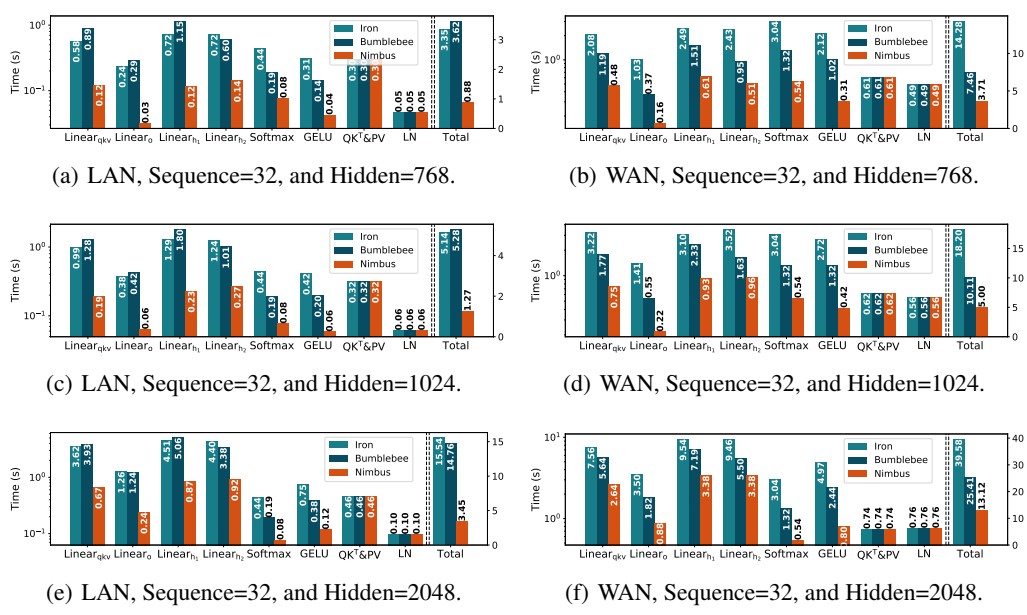

(a) LAN, Sequence=32, and Hidden=768.

(b) WAN, Sequence=32, and Hidden=768.

(c) LAN, Sequence=32, and Hidden=1024.

(d) WAN, Sequence=32, and Hidden=1024.

(e) LAN, Sequence=32, and Hidden=2048.

(f) WAN, Sequence=32, and Hidden=2048.

Figure 9: Under sequence length 32, the end-to-end speedup and breakdown of varying hidden dimensions and network conditions.

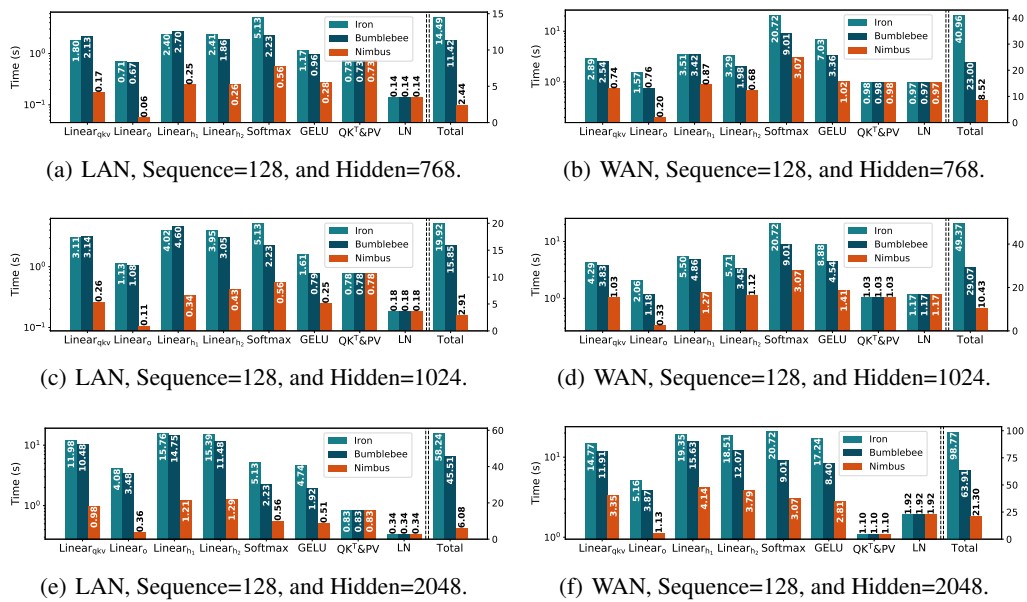

(a) LAN, Sequence=128, and Hidden=768.

(b) WAN, Sequence=128, and Hidden=768.

(c) LAN, Sequence=128, and Hidden=1024.

(d) WAN, Sequence=128, and Hidden=1024.

(e) LAN, Sequence=128, and Hidden=2048.

(f) WAN, Sequence=128, and Hidden=2048.

Figure 10: Under sequence length 128, the end-to-end speedup and breakdown of varying hidden dimensions and network conditions.

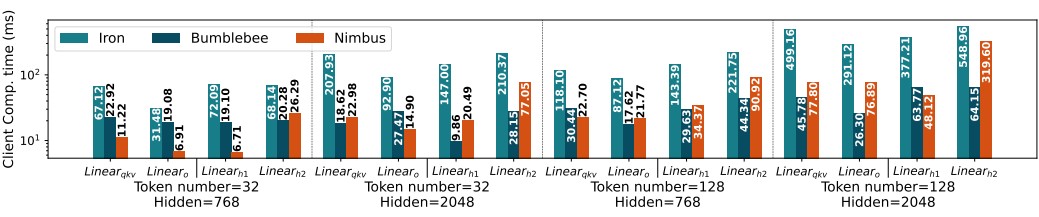

Figure 11: Under different sequence lengths and hidden sizes, we present comprehensive experiments of client computation time of different methods.

directly comparing computation complexity is hard to draw a conclusion due to the choice of the window size, we profile the client's computation time to compare the computation workload.

Under different sequence lengths and hidden sizes, Figure 11 presents comprehensive experiments of client computation time of different methods. Across varying model sizes and input sequence lengths, Nimbus only takes around 20% to 30% compared with Iron and $0.7\times$ to $2.7\times$ compared with BumbleBee. The BumbleBee has less client computation because the compression used in BumbleBee allows the client to encrypt her activation shares to less number of ciphertexts and receive fewer output ciphertexts, thereby requiring fewer NTT operations. In the worst case of the $\text{Linear}_{h2}$, the client computation is around $5.0\times$ longer than the BumbleBee. But still, on the total Transformer block, the extra overhead is only $2.7\times$. This extra computation ratio can be further shrunk given the base that the client needs to perform a large amount of computation of non-linear layers. Therefore, we believe this is acceptable for a powerful client in MPC.

**Asynchronous Weights Loading.** For varying hidden dimensions, Table 4 lists the encrypted weight size and corresponding loading time from disk to memory, which usually takes less than 1 second. For the hidden size 768, which is the size of $\text{BERT}_{\text{base}}$ mainly considered in this paper. The loading time is only 90 ms and 370 ms, which can be easily overlapped by later communication. Therefore, we can only keep a limited number of encrypted weights in the memory, such as weights of four linear layers of a Transformer block. Then, we swap the later weights during the execution.

Table 4: The size of the encrypted weights measured by megabytes (MB) and the corresponding loading time measured by seconds (s).

| Hidden | Linear$_{qkv}$ | | Linear$_o$ | | Linear$_{h1}$ | | Linear$_{h2}$ | |
|---|---|---|---|---|---|---|---|---|
| | Size | Time | Size | Time | Size | Time | Size | Time |
| 768 | 180 | 0.09 | 180 | 0.09 | 180 | 0.09 | 720 | 0.37 |
| 1024 | 240 | 0.13 | 240 | 0.13 | 240 | 0.13 | 960 | 0.50 |
| 2048 | 480 | 0.25 | 480 | 0.25 | 480 | 0.25 | 1920 | 0.91 |

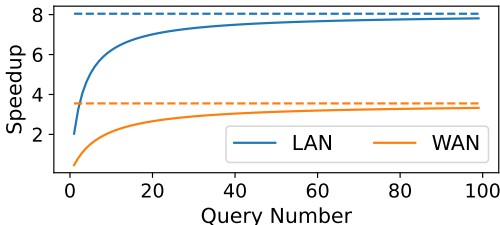

Figure 12: The execution+setup speedup over BumbleBee under different queries.

### F.3 Amortized Overhead of the encrypted weights.

Our linear protocol replaces the input communication with a one-time setup communication of sending encrypted weights. Although our method focuses on the overhead of the execution phase, we are also interested in how many queries can make the amortized setup overhead negligible. For BERT$_{base}$, using our row-wise encoding, the encrypted weight size of four types of layer within a Transformer block is 180MB, 180MB, 180MB, and 720MB. We incorporate the one-time overhead of transmitting these weights into our analysis and amortize it across the number of queries. Figure 12 shows the speedup of amortized Nimbus over BumbleBee. The dot lines indicate the speedup without considering the setup overhead. There are two critical points that we are interested in. At around three queries, the amortized overhead begins to surpass the BumbleBee, leading to a speedup greater than 1. At around 40 queries, the maximal speedup is achieved.

### F.4 More Accuracy and Efficiency Comparisons of Non-linear Approximations

Besides BumbleBee [26], this section compares non-linear approximation with other state-of-the-art (SOTA) works, including MPCFormer [22] and BOLT [29].

**Accuracy Comparison** The details of the accuracy are listed in Table 5. The accuracy of MPCFormer is reproduced using the open-sourced code of them. MPCFormer's idea is to adopt aggressive approximation to tailor the efficiency of the MPC but sacrifice the inference accuracy. For example, they let $GELU(x) = 0.125x^2 + 0.25x + 0.5$. MPCFormer can efficiently compute the Softmax and GELU. However, even using knowledge distillation to recover the accuracy, they still incur 2.26% accuracy on average. BOLT optimizes exponential and GELU using integer-only approximation and

Table 5: Accuracy comparison of floating-point (FP) baseline, BumbleBee, MPCFormer (reproduced using Quad+2ReLU), BOLT$^*$ (accuracy relative change from the original paper), Nimbus (with finetuning). We report the relative change in accuracy.

| Method | CoLA | SST-2 | MRPC | STS-B | QQP | MNLI | QNLI | RTE | Avg. |
|---|---|---|---|---|---|---|---|---|---|
| | Matthews corr. | Acc. | F1 | Pearson | Acc. | Acc. | Acc. | Acc. | |
| FP baseline | 58.63 | 92.66 | 90.12 | 88.24 | 91.22 | 84.74 | 91.28 | 67.87 | 83.10 |
| Bumblebee | -0.23 | 0.00 | 0.00 | 0.04 | -0.01 | 0.00 | 0.11 | 0.00 | -0.01 |
| MPCFormer | -5.88 | -1.04 | -0.73 | -3.03 | -1.91 | -1.42 | -0.51 | -3.57 | -2.26 |
| BOLT | - | -0.62 | 0.53 | -1.65 | - | - | - | -0.02 | -0.44 |
| Nimbus | -0.23 | -0.11 | 0.30 | -0.12 | -0.24 | -0.37 | 0.09 | 0.00 | -0.08 |

Table 6: Efficiency comparison of BumbleBee, MPCFormer (Quad+2ReLU), BOLT, Nimbus

| Method | LAN (3Gbps, 0.8ms) | | WAN (200Mbps, 40ms) | |
|---|---|---|---|---|
| | GLUE | Softmax | GLUE | Softmax |
| Bumblebee | 0.96 | 2.13 | 9.16 | 19.87 |
| MPCFormer | 0.08 | 0.71 | 1.16 | 7.05 |
| BOLT | 3.18 | 3.66 | 22.20 | 23.16 |
| Nimbus | 0.28 | 0.56 | 2.60 | 5.86 |

piecewise polynomial approximation, respectively. Similar to the BumbleBee, their approximation regards all input values as equal importance. Therefore, they require a relatively high-degree polynomial to approximate the original function. Since BOLT does not opensource the codes used to fine-tune the model, we use accuracy numbers reported in their paper. Their approximations are relatively accurate and incur 0.44% average accuracy loss after fine-tuning.

**Efficiency Comparison** The efficiency comparisons for LAN and WAN settings are listed in Table 6. We utilize the open-sourced code of BOLT to reproduce their latency. MPCFormer is originally implemented on the Crypten framework [20], which uses secret sharing as its underlying cryptographic primitive instead of homomorphic encryption. To facilitate a fair comparison of the effectiveness of non-linear approximation, we re-implement their polynomial approximation using SecretFlow as the backend.

MPCFormer's GELU does not require comparison and is extremely fast. They use ReLU to replace the exponential in the Softmax, which is also fast. The same as the ReLU substitution of MPCFormer, Nimbus's approximation of the exponential also has one comparison, and the additional three multiplications are relatively quick to compute. Furthermore, our approach allows for computation on a smaller ring, which makes us faster than MPCFormer's Softmax. Our method is approximately $10\times$ faster than BOLT in all cases. In addition to the advantages of our simpler approximation, other factors also contribute to the slowness of BOLT. First, BOLT's linear layer is evaluated in the field, necessitating extra operations to convert field elements to ring elements. Second, our simpler approximation reduces the fixed-point error during computation, enabling computation on a smaller ring. Since other layers still require computations on a larger ring, we also propose a truncation-upcast fusion protocol to eliminate the overhead of upcasting ring elements from a smaller ring to a larger one. In contrast, BOLT requires additional communication for this upcasting process.

