# OpenReview forum: "Nimbus: Secure and Efficient Two-Party Inference for Transformers"
_NeurIPS.cc/2024/Conference — NeurIPS 2024 poster_

### Official Review · Reviewer_zib5 · 2024-07-08

**Soundness:** 3
**Presentation:** 3
**Contribution:** 3
**Rating:** 6
**Confidence:** 4

**Summary:**

This work proposes a new secure two-party computation (2PC) protocol called Nimbus for Transformer models to improve the efficiency and effectiveness of large matrix multiplication and non-linear layer approximation in Transformer inference. First, this work exploits client-side outer product and output compact to enhance layer multiplication. Second, input distributions are taken into consideration to better approximate GELU and exponential with lower-order piecewise polynomials. Comprehensive experiments and analyses are demonstrated in this work, and the results indicate Nimbus is effective and efficient.

**Strengths:**

- Nimbus proposes client-side outer product (COP) and output compact with shift operation to reduce the computing and communication overhead.

- Nimbus considers the impact of input distribution to simplify the polynomial approximation of GELU and Softmax with lower-order piecewise polynomials and small rings.

- The authors provide a comprehensive discussion of Nimbus efficiency and feasibility, including client-side resources, asynchronous weight loading, free ring conversion, and more.

- This work presents comprehensive complexity analyses, protocol definitions, and evaluation experiments, making the results and conclusion convincing and practical.

- The evaluations prove that Nimbus significantly improves the performance and efficiency of secure 2PC for Transformer models, compared with existing works like BumbleBee, Iron, and BOLT.

**Weaknesses:**

The major concerns are the accuracy and feasibility of estimating input distribution by sampling.

- The server samples a batch of data from the training dataset to estimate the input distribution. I notice the work lack details of this process. For example, what is the exact batch size of sampling? I think such hyper-parameter can influence the effectiveness and efficiency of sampling. Does Nimbus compute the results of all training data to get the distribution, or just sample a subset? The precise process of sampling is worth further mentioning.

- It seems that Nimbus assumes that the training data and test data share the same distribution, or at least the distributions are similar. What if the two distributions are more significantly skewed?

- The initial state of piecewise approximation polynomials is pre-defined empirically despite the precise split points optimized by equation (3). For example, the approximation for GELU is divided into three pieces, with the middle piece being a quadratic polynomial. However, what if the distributions of inputs differ in practical cases? Figure 4 illustrates the distribution of non-linear functions for one specific dataset. If there exists another dataset with a more uniform distribution, such pre-defined piecewise polynomials may result in unwanted inaccuracy.

- In some special cases, to further protect privacy, not only the inference stage is protected by secure 2PC, but the training dataset are also protected by secure training. In such case, the server may fail to estimate the input distributions.

**Questions:**

Please refer to the contents of the Weaknesses part.

Besides,  there are some writing mistakes. For example, in Line 4 of Algorithm 1 in the appendix, it should be $\widetilde{c} - r(\theta)$ rather than $r(\theta) - \widetilde{c}$.

**Limitations:**

As mentioned above, the input distributions estimated by sampling when approximating non-linear functions may suffer from inaccuracy, and the description of the exact sampling process is not very clear.

---

> ### Author Rebuttal · Authors · 2024-08-06
>
> We thank the reviewer for your helpful comments, and address your concerns as follows. We also appreciate the reviewer's attentive reading for pointing out the typo in the Appendix. We will rectify this in the revised version.
> # Q1: Details of the batch size for summarizing the input distribution
> For all eight tasks in Table 2, we randomly sample sentences from the training dataset until the total token count reaches 512. This is based on the finding that the distribution of intermediate activations stabilizes when the sampled data exceeds 256 tokens. We have included figures illustrating this observation in Figure 2 of the **supplementary PDF** under the "global" response.
>
> # Q2: What if distribution of the test data are significantly skewed from the training data
> The distribution skewness between the training and test data indeed exists and causes the generalization error of the model's accuracy. Nimbus is also affected by this skewness. As shown in Table 2, the skewed data contributes to the accuracy loss in the third line. However, the skewness does not have a significant impact and can be recovered through lightweight fine-tuning. The skewness is less significant for DNN models since the accuracy of DNN models relies on the independent and identically distributed (i.i.d) assumption among the training and test datasets [1], and many techniques are proposed to guarantee this assumption. For example, normalizing the training and test data using the same statistics, and using layer normalization to make the model more robust to distribution changes. Pretraining Transformers requires a large amount of data so that the model is trained to map data to the same hidden space. Many model compression techniques also utilize this assumption to design quantization [2] or pruning [3] strategies. Nimbus also builds on such a common assumption.
>
> [1] Learning to Reason with Neural Networks: Generalization, Unseen Data and Boolean Measures, NIPS 2022
>
> [2] Deja vu: Contextual sparsity for efficient llms at inference time, ICML 2023
>
> [3] Smoothquant: Accurate and efficient post-training quantization for large language models, ICML 2023
>
> # Q3: Generalization of the insight on activation distribution to other datasets
> We supplement our study with additional experiments on the activation distribution (Figure 1 in the **supplementary PDF** of the global response) to demonstrate the generalization. Besides the observations provided in the paper, we verify the activation distribution on additional popular datasets. Our experiments include the BERT-base model on the MRPC dataset (for sequence classification), SQuAD dataset (for question & answering), and SWAG dataset (for multiple choice), along with the GPT-2 model on the Wikitext-2 dataset (for causal language modeling). We observe obvious non-uniform distributions across these datasets. Furthermore, these distributions exhibited similar patterns, indicating that the piece splitting strategy proposed in this paper can be directly applied to other datasets. The regular distribution of intermediate activations has also been verified by prior works as a widely applicable rule across various Transformer models and tasks [1,2], where it has been utilized for quantization and sparsity in [1,2].
>
> Therefore, compared to the previous strategy of treating the input distribution as uniform, our distribution-aware fitting is expected to yield better fitting results. We also conduct accuracy experiments on these datasets. As the following table shows, Nimbus only has a minor impact on the accuracy.
> | Method      | BERT-base (MRPC) | BERT-base (SQuAD) | BERT-base (SWAG) | GPT2 (wikitext-2)  |
> |-------------|------------------|-------------------|------------------|--------------------|
> |             | F1               | F1                | accuracy         | perplexity         |
> | FP baseline | 90.12            | 88.1              | 81.08            | 20.01              |
> | Nimbus      | 90.42            | 87.93             | 80.94            | 21.36              |
>
>
> [1] Deja vu: Contextual sparsity for efficient llms at inference time, ICML 2023
>
> [2] Smoothquant: Accurate and efficient post-training quantization for large language models, ICML 2023
>
>
> # Q4: Estimation of input distribution when the training dataset is also secret for the server
> This case poses a greater challenge for secure inference and is interesting to explore. In this scenario, the intermediate activations are invisible to the server so that he cannot directly estimate the distribution. The server can employ privacy-computing techniques such as MPC and differential privacy to estimate the activation distribution, which is feasible even though the training dataset is secret. As mentioned in question 1, our method only requires a small batch of data (e.g., 512 tokens in total) to estimate the distribution. Because the data volume is small and the estimation is a one-time setup task, the performance using a privacy-computing technique to estimate the activation distribution is acceptable. We believe that our solution will remain attractive for such a more challenging case.

---

### Official Review · Reviewer_7qmF · 2024-07-12

**Soundness:** 2
**Presentation:** 2
**Contribution:** 2
**Rating:** 5
**Confidence:** 4

**Summary:**

This paper proposes Nimbus, a secure inference protocol for transformers in the 2pc setting. They propose distribution-aware nonlinear function approximation to use low-degree polynomials to compute GELU and softmax. They showed that their method can preserve accuracy and achieve efficient performance by comparing it with several baselines.

**Strengths:**

- Improving the efficiency of the secure inference systems is an important and timely topic.
- Extensive experiments are conducted to demonstrate the efficiency and accuracy of the proposed system.

**Weaknesses:**

- The nonlinear approximation is leaking private information.
- This proposed system might require huge storage overhead on the client side.
- There is incorrect information regarding BOLT (S&P'24) and BumbleBee (NDSS'24).
- There is a gap between the authors' reported results and BumbleBee's results in their paper, which needs further explanations.

**Questions:**

- The major reason that prevents me from advocating this paper is that the nonlinear function approximations actually leak sensitive information about the training input distribution, which is not desirable in a secure inference system. For instance, the approximations polynomials are available to both parties, such that the client can easily reverse engineering the training inputs' distribution. For the GELU function, you can save one comparison and multiplication in 2PC compared to Bumblebee and BOLT but at the cost of leaking private information.
- Additionally, the input ranges for different models, datasets, training parameters, and even different layers could be significantly different. Thus, you might need different approximations for different tasks, which will leak model information and is not friendly to use.
- In your system, the matrix multiplication is done on the client side, which introduces a large memory overhead on the client side as they need to load the encrypted model into the memory. The encrypted model could be hundreds of times larger than the plaintext model. Considering that the client could be a normal user, such an issue should be avoided in a 2PC inference system.
- I'm wondering why BumbleBee's performance is not as good as reported in their paper. It's about 5x faster compared to your reported numbers. Are you running their code correctly?
- In line 308, you mentioned that BOLT uses aggressive approximation for efficiency, which is not the case. MPCFormer indeed introduces significant modifications to the nonlinear functions, but I believe BOLT's approximations are accurate, and they should be comparable to your designs. Additionally, BOLT seems to be open-sourced as BumbleBee mentioned that they obtained the results by rerunning their code.

**Limitations:**

See my concerns in the Question section.

---

> ### Author Rebuttal · Authors · 2024-08-06
>
> Thank you for your thoughtful feedback on our paper. We appreciate your insights and would like to provide more clarification.
> # Q1, Q2: The leakage of the approximation polynomial
> The secure polynomial evaluation does not allow the client to learn the polynomial coefficients and comparison thresholds, except that the coefficient on the highest degree term is known by the client to achieve slightly better efficiency. It is easy to eliminate the leakage of the high-degree coefficient by introducing an overhead of about 8%. See below for more details.
>
> We use the GELU function as an example to show how securely computing a polynomial keeps the coefficients and thresholds secret from the client. We first discuss the coefficients. Consider the approximation polynomial $P^2(x)=b_0 x^2 + b_1 x + b_2$. The client and server hold an additive secret sharing on the input secret $x$ denoted by $\langle x \rangle = (x_c, x_s)$, and only the server knows $(b_0, b_1, b_2)$. To compute the additive secret sharing of $P^2(x)$, two parties first jointly compute $\langle A_1 \rangle = \Pi_{mul}(\langle b_0 \rangle \cdot \langle x \rangle)+b_1$, where $\Pi_{mul}$ is the secure multiplication technique in BumbleBee [29]. Then two parties jointly compute $\langle A_2 \rangle = \Pi_{mul}(\langle A_1 \rangle \cdot \langle x \rangle)+b_2$ as the sharing of $P^2(x)$. During the evaluation, $b_0$ is kept secret through secret sharing, while $b_1$ and $b_2$ can be added locally at the server side using the linear property of additive secret sharings. This ensures that $b_0, b_1, b_2$ remain secret from the client. Second, for the comparison thresholds. To compare $x$ with a secret threshold $T$ known only by the server, the server can locally compute the additive secret sharing $\langle x-T \rangle = \langle x \rangle - T$ using the linear property of additive secret sharings. Then, both parties execute a secure comparison protocol on the input $\langle x-T \rangle$ to obtain an additive secret sharing of the bit indicating whether $x-T$ is greater than zero or not. Thus, the threshold $T$ is kept secret.
>
> In this paper, to achieve better performance, we choose to let the server send $b_0$ to the client. As illustrated in line 1 of Alg. 3, this can eliminate one call to $\Pi_{mul}(\langle b_0 \rangle \cdot \langle x \rangle)$ and instead compute $b_0 \cdot \langle x \rangle$ locally. This is a trade-off between security and efficiency. However, it is important to note that simply revealing the highest degree coefficient is not sufficient for the client to recover the input distribution. If the leakage is still a privacy concern, we can keep the highest-degree coefficient $b_0$ secret as the protocol in the above paragraph mentioned. We denote Nimbus* as the slightly modified protocol in the above paragraph that keeps $b_0$ secret and guarantees zero leakage on the polynomial. Following the same experimental setup, Nimbus* only increases the overhead by around 8% for secure computation of non-linear functions compared to Nimbus that reveals $b_0$. See the following table for the efficiency comparison, where the values are measured in seconds.
> |Method| GLUE (LAN) | Softmax (LAN) | GLUE (WAN)| Softmax (WAN)|
> |-|-|-|-|-|
> |Bumblebee|1.17|2.23|3.36|9.01|
> |Nimbus|0.28|0.56|1.02|3.07|
> |Nimbus*|0.30|0.61|1.10|3.29|
>
> # Q3: The memory concern of loading encrypted model into the memory
> We discussed this concern in Section 3.3. The encrypted model weights are expanded to at least four times the size of the plaintext weights. However, our solution does not store all the ciphertexts on model weights in memory. Instead, we only load the ciphertexts for a limited number of layers into memory. This approach is based on the insight that secure inference is primarily bottlenecked by network communication. By overlapping the swapping of ciphertexts from local disk to memory with network communication, we add no extra running time to load a part of ciphertexts on model weights. See Section 3.3 for more details.
>
> # Q4: Performance gap between the reported results and BumbleBee’s results in their paper
> Our experiments are based on the open-source codes of BumbleBee and are expected to produce similar results to theirs. We have double-checked the experimental results reported in this paper against those reported in BumbleBee and find similar performance results. Note that while BumbleBee reports the performance of the whole model, we report the performance of each layer.
>
> For example, we consider the experiment over a WAN, using almost identical settings as BumbleBee: a 400Mbps network bandwidth, the BERT-base model, and an input length of 128. In Table V of BumbleBee, they state that the 12-layer model takes 4.86 minutes, which translates to approximately 24.3 seconds per layer. This is very close to the 23 seconds per layer reported in our Table 6(b). Despite differences in hardware, the performance data from a concrete implementation is valid. We are happy to provide source codes for experimental reproduction.
>
>
> # Q5: Comparison with BOLT's nonlinear approximation
> We say that BOLT uses aggressive approximation, meaning that the numbers reported in BOLT show an accuracy loss, as shown in Table 2 of BOLT's paper. We use the term "aggressive" to differentiate it from the approximation with (almost) no accuracy loss used in Iron and BumbleBee. Indeed, BOLT's approximation has a much smaller impact on accuracy, compared to MPCFormer. We will clarify it, and instead say that BOLT has a larger accuracy loss than Iron, BumbleBee and Nimbus, in the updated version.
>
> By the time we submitted the paper, BOLT had not released their codes. Therefore, we implemented their nonlinear approximation using the backend of SecretFlow and compared the performance in Appendix G.4. Here, we also include performance comparison using their official codes, which can be found in Table 1 of the **supplementary PDF** under the "global" response.

---

> > ### Comment · Reviewer_7qmF · 2024-08-08
> >
> > Thanks for the authors' rebuttal. I have comments as follows:
> >
> > Thanks for the clarifications on the function privacy. I think it should be clearly explained in the paper.
> >
> > I still do not understand where are the savings from. In your GELU approximation, you need to evaluate 2 multiplications between secret shares and 2 comparisons, which should be the same as BOLT. Are the savings from the efficient crypto primitives used in Bumblebee?
> >
> > Given that my major concern is addressed, I've slightly raised my score. I suggest the authors add a paragraph to discuss the function privacy in the revision.

---

> ### Author Response · Authors · 2024-08-09
>
> We are pleased to hear that the rebuttal addresses your concerns and thank you for raising your score. We will clarify the security of the polynomial approximation protocol in the revision. We will also describe the zero-leakge version of the protocol, and provide the performance comparison.
>
> For the question of where the savings come from, there are three advantages in theoretical to consider. Firstly, the number of truncations called in Nimbus is the same as the number of secure multiplications called, which is two. In contrast, BOLT calls secure multiplication twice but makes four truncation calls. This is because BOLT includes additional multiplication between the public value and secret. While the multiplication between the public value and secret can be done locally, the truncation of the result sharing cannot be saved. The truncation protocol used in BOLT requires $\log \ell+3$ rounds of communication [1], where $\ell$ represents the bit length of the ring. As a result, this increases the overall communication overhead. Secondly, our low-degree approximation reduces the fixed-point error during computation, allowing computation on a smaller ring. Since other layers still require computation on a higher ring, to eliminate the overhead of upcasting ring elements from a smaller ring to a larger ring, we also propose a truncation-upcast fusion protocol. In contrast, BOLT requires $\log m+2$ rounds of communication [1] for this upcast, where $m$ is the bit length of the smaller ring. Thirdly, all computations of the Nimbus are performed on the ring, while the BOLT evaluates the linear layer on the field and the nonlinear layer on the ring. As a result, an additional conversion between the field and the ring is needed. As for the implementation, Nimbus uses HE based secure multiplication [2], while the BOLT uses OT-based secure multiplication [1]. The former saves more communication, and performances better when the network condition is bad. The underlying OT protocol also causes difference. The SOTA Ferret OT [3] used in Nimbus takes less communication than the IKNP OT [4] used by BOLT.
>
> [1] Sirnn: A math library for secure rnn inference, SP 2021
> [2] BumbleBee: Secure Two-party Inference Framework for Large Transformers, NDSS 2021
> [3] Ferret: Fast extension for correlated ot with small communication, CCS 2020
> [4] Extending Oblivious Transfers Efficiently, CRYPTO 2003

---

### Official Review · Reviewer_5SDb · 2024-07-12

**Soundness:** 3
**Presentation:** 3
**Contribution:** 3
**Rating:** 6
**Confidence:** 3

**Summary:**

This paper provides a hybrid method that uses both HE and additive secret sharing (Add-SS) to perform 2PC privacy-preserving transformers. Two main contributions are discussed in this paper: (1) Client-side Outer Product Protocol and (2) Lower Degree Polynomial Approximation and Smaller Rings.

**Strengths:**

The paper presents the complexity analysis and memory impact analysis well. These analyses help readers understand the advantages of the COP protocol.

**Weaknesses:**

1. Not quite sure why HE + Add-SS is used for 2PC; why not directly use multi-party computing for 2PC? Please compare the differences between these two techniques and related papers, as it is uncertain whether HE + Add-SS is a more promising technique for privacy-preserving transformers.
2. The Client-side Outer Product Protocol is one of the main contributions. However, there is no security proof for section 3.2 Client-side Outer Product Protocol, especially for the content from lines 165 to 170.

**Questions:**

Why HE + Add-SS is used for 2PC; why not directly use multi-party computing for 2PC?

**Limitations:**

No security proof for section 3.2 Client-side Outer Product Protocol

---

> ### Author Rebuttal · Authors · 2024-08-06
>
> We thank the reviewer for your helpful comments, and would like to address the main concerns as follows.
> # Q1: Why HE + Add-SS is used for secure two-party inference
> HE combined with additive secret sharing (Add-SS) is one of the most promising techniques for secure two-party DNN inference. The technique of HE+Add-SS has been widely used in prior works such as Gazelle [17], Delphi [38], Cheetah [16], Iron [13], BOLT [33], and BumbleBee [29]. The rationale behind this is that HE allows the communication cost of securely computing linear layers to be independent of the model parameters. While other 2PC approaches such as GMW, Beaver, and garbled circuit (GC) require communication linear in the size of model parameters, the HE approach achieves significantly lower communication for linear layers. For non-linear layers, the 2PC protocol mainly adopts the Add-SS approach, which achieves the best efficiency for now. In particular, Add-SS, which is often combined with the oblivious transfer (OT) extension protocol, is particularly suitable for securely computing Boolean circuits, which are currently the most efficient circuit representation for non-linear functions. The Add-SS approach also obtains much lower communication than other approaches such as GC. We will clarify this in the updated version.
>
> # Q2: Security proof of the client-side outer product protocol
> We omit the security proof since the security of our client-side outer product protocol directly builds upon a secure homomorphic encryption (HE) scheme, and thus its security proof is somewhat straightforward. In particular, our protocol guarantees the same security as the traditional server-side inner product protocol. In the presence of a semi-honest adversary, we provide a brief proof idea below, and will include a formal proof in the Appendix.
>
> Specifically, the model weights are encrypted by the server and then sent to the client, where the ciphertexts are denoted by Enc(W). From the CPA security property of the HE scheme, the ciphertexts reveal no information about these model weights. For secure matrix multiplication, the input matrix X has been shared as $(X_c, X_s)$ using Add-SS. The client samples a matrix of random shares $Y_c$, and then homomorphically computes a ciphertext $Enc(X_c * W-Y_c)=X_c * Enc(W)-Y_c$. Due to the circuit-privacy property of the HE scheme, in the client view, the ciphertext $Enc(X_c * W-Y_c)$ does not reveal information on $X_c$ and $Y_c$. The ciphertext $Enc(X_c * W-Y_c)$ is sent to the server, who decrypts it to a matrix of shares $T_s$ such that $Y_c$ and $T_s$ constitute additive sharings of $X_c * W$. Finally, the server can locally compute its shares $Y_s=T_s+X_s * W=X * W-Y_c$, and $(Y_c, Y_s)$ constitutes the additive sharings of $X * W$. It is natural to see that the local computation is secure. In the proof of security, the simulator can simulate the HE ciphertexts using "dummy" ciphertexts on zero, and the adversary's view between the real-world execution and ideal-world execution is proven to be computationally indistinguishable by reducing it to the CPA and circuit-privacy security of the HE scheme.

---

### Official Review · Reviewer_dGsE · 2024-07-15

**Soundness:** 2
**Presentation:** 3
**Contribution:** 2
**Rating:** 5
**Confidence:** 4

**Summary:**

This submission proposed secure inference protocols for Transformer-based model, involving two
crucial components: HE-based linear operations and approximation-based no-linear operations.
Experiments were conducted to verify the feasibility of the proposed protocols and to compare the
performance with prior works on Transformer secure inference.

**Strengths:**

The author focused on the critical part of designing efficient MPC protocols for Transformer, with
detailed analysis on drawbacks of prior works.

For the linear layer, the solution takes into consideration the different computation sources that
the client and server hold. As such, during the linear operations, the server side is responsible for
the heavy cryptographic operations.

For the non-linear layer, the protocol in the submission is based on the distribution of function input.
Concretely, low-degree approximation was used, for the selected range.
The experiment in the paper is comprehensive and complete. Results indicate that it outperforms the
baseline.

**Weaknesses:**

1. One of the insights of the non-linear construction is based on the non-uniformed distribution of the input. It is not quite convincing as insufficient explanation was provided, thus limiting the feasibility of the protocol. Is it applicable to all types of input datasets or just limited to certain categories?

2. The contributions and novelty of the non-linear design are not sufficiently highlighted. It can be
viewed as the variant of spline approximations with carefully selected coefficients, which is
commonly applied in prior works [1].

3. Truncation in Alg. 5 discards the high-order bits rather than low-order bits. However, commonly
it is the variant of the logical right shift [2]. Please clarify the definition of such an operation.

[1] Hou, Xiaoyang et al. “CipherGPT: Secure Two-Party GPT Inference.” IACR Cryptol. ePrint
Arch. 2023 (2023): 1147.

[2] D. Rathee et al., "SiRnn: A Math Library for Secure RNN Inference," 2021 IEEE Symposium
on Security and Privacy (SP), San Francisco, CA, USA, 2021, pp. 1003-1020,

**Questions:**

See above.

**Limitations:**

See above.

---

> ### Author Rebuttal · Authors · 2024-08-06
>
> We thank the reviewer for your insightful feedback and suggestion. We address the main concerns as follows.
> # Q1: Generalization of insight on activation distribution to other datasets
> We supplement our study with additional experiments on the activation distribution (Figure 1 in the **supplementary PDF** of the global response) to demonstrate the generalization. Besides the eight tasks in the GLUE benchmark provided in the paper, we verify the activation distribution on additional popular datasets. Our experiments include BERT-base on the MRPC dataset (for sequence classification), SQuAD dataset (for question & answering), and SWAG dataset (for multiple choice), GPT-2 on the Wikitext-2 dataset (for causal language modeling). We observe obvious non-uniform distributions across these datasets. These distributions also exhibit similar patterns, indicating that the piece splitting strategy proposed in this paper can even be directly applied to other datasets. The non-uniform distribution of intermediate activations has also been verified by other studies as a widely applicable rule across various Transformer models and tasks [1,2], which they have utilized for quantization and sparsity.
>
> Therefore, compared to the previous strategy that treats the input distribution as uniform, our approach of fitting nonlinear functions according to the activation distribution is expected to yield better fitting results. We also conduct accuracy experiments on these datasets. As the following table shows, Nimbus only has a minor impact on the accuracy.
> | Method      | BERT-base (MRPC) | BERT-base (SQuAD) | BERT-base (SWAG) | GPT2 (wikitext-2)  |
> |-------------|------------------|-------------------|------------------|--------------------|
> |             | F1               | F1                | accuracy         | perplexity         |
> | FP baseline | 90.12            | 88.1              | 81.08            | 20.01              |
> | Nimbus      | 90.42            | 87.93             | 80.94            | 21.36              |
>
>
> [1] Deja vu: Contextual sparsity for efficient llms at inference time, ICML 2023
>
> [2] Smoothquant: Accurate and efficient post-training quantization for large language models, ICML 2023
>
>
>
>
>
> # Q2: Clarification of the contributions on nonlinear layers
> Our work on nonlinear layers has two main contributions. First, we observe that the activation distribution of the transformer model is non-uniform. Based on this observation, we allocate the approximation budget based on the input distribution when fitting the nonlinear function. This is different from prior works that directly minimize the output difference between the original function and approximated polynomials, assuming a uniform input distribution. Second, we have found that our low-degree approximation allows for efficient small ring computation. To support this, we also propose a truncation-upcast fusion protocol to avoid the cost of ring upcast. We will explain our twofold contributions in more detail in a later revision.
>
>
> # Q3: Clarification of truncation
> Our truncation in Alg. 5 also discards the low-order bits. We give a detailed derivation of Alg. 5 in Equation (8) of the Appendix. You can check the last line of Equation (8), where the low-order bits of $\langle x \rangle$ are discarded by division by $2^s$. We will further clarify this in the updated version of Alg. 5. We are open to further discussion if there are any remaining questions.

---

### Author Rebuttal · Authors · 2024-08-06

# Global response
Thank you for taking the time to review our work. Besides the separate response, we also include a **PDF** file under the global response that contains figures and a table related to the reviewers' comments. If you have any further questions or need more details, please don't hesitate to reach out. Your feedback is important to us, and we are ready to explain anything further. Thanks again for your comments!

---

### Decision · Program_Chairs · 2024-09-25

**Decision:**

Accept (poster)

**Comment:**

All reviewers agreed this paper should be accepted: it addresses an important problem, the experiments are comprehensive, and the distribution-aware approximation is interesting. Authors: you've already indicated that you've updated the submission to respond to reviewer changes, if you could double check their comments for any recommendation you may have missed on accident that would be great! The paper will make a great contribution to the conference!